# Regulation of inflammation and protection against invasive pneumococcal infection by the long pentraxin PTX3

Rémi Porte[1,2]*[†], Rita Silva-Gomes[1,2†‡], Charlotte Theroude[3], Raffaella Parente[1], Fatemeh Asgari[1], Marina Sironi[1], Fabio Pasqualini[1,2], Sonia Valentino[1], Rosanna Asselta[1,2], Camilla Recordati[4,5], Marta Noemi Monari[1], Andrea Doni[1], Antonio Inforzato[1,2], Carlos Rodriguez-Gallego[6], Ignacio Obando[7], Elena Colino[8], Barbara Bottazzi[1]*, Alberto Mantovani[1,2,9]*

[1]IRCCS Humanitas Research Hospital, Milan, Italy; [2]Department of Biomedical Sciences, Humanitas University, Milan, Italy; [3]Infectious Diseases Service Laboratory, Department of Medicine, Lausanne University Hospital, University Hospital of Lausanne, Lausanne, Switzerland; [4]Mouse and Animal Pathology Laboratory, Fondazione Filarete, Milan, Italy; [5]Department of Veterinary Medicine, University of Milan, Lodi, Italy; [6]Department of Clinical Sciences, University Fernando Pessoa Canarias, Las Palmas de Gran Canaria, Spain; [7]Department of Pediatrics, Hospital Universitario Virgen del Rocío, Sevilla, Spain; [8]Department of Pediatrics, Complejo Hospitalario Universitario Insular Materno Infantil, Las Palmas de Gran Canaria, Spain; [9]William Harvey Research Institute, Queen Mary University of London, London, United Kingdom

**\*For correspondence:**
remi.porte@inserm.fr (RP); Barbara.Bottazzi@humanitasresearch.it (BB); Alberto.Mantovani@humanitasresearch.it (AM)

[†]These authors contributed equally to this work

**Present address:** [‡]Life and Health Sciences Research Institute (ICVS), School of Medicine, University of Minho and ICVS/3B's - PT Government Associate Laboratory, Braga, Portugal

**Abstract** *Streptococcus pneumoniae* is a major pathogen in children, elderly subjects, and immunodeficient patients. Pentraxin 3 (PTX3) is a fluid-phase pattern recognition molecule (PRM) involved in resistance to selected microbial agents and in regulation of inflammation. The present study was designed to assess the role of PTX3 in invasive pneumococcal infection. In a murine model of invasive pneumococcal infection, PTX3 was strongly induced in non-hematopoietic (particularly, endothelial) cells. The IL-1β/MyD88 axis played a major role in regulation of the *Ptx3* gene expression. *Ptx3*[−/−] mice presented more severe invasive pneumococcal infection. Although high concentrations of PTX3 had opsonic activity in vitro, no evidence of PTX3-enhanced phagocytosis was obtained in vivo. In contrast, *Ptx3*-deficient mice showed enhanced recruitment of neutrophils and inflammation. Using *P-selectin*-deficient mice, we found that protection against pneumococcus was dependent upon PTX3-mediated regulation of neutrophil inflammation. In humans, *PTX3* gene polymorphisms were associated with invasive pneumococcal infections. Thus, this fluid-phase PRM plays an important role in tuning inflammation and resistance against invasive pneumococcal infection.

## Editor's evaluation

This submission represents a holistic approach to how pentraxin 3 (PTX3) modulates susceptibility to experimental infection by Streptococcus pneumoniae. The authors have built robust findings on the importance of PTX3 for the survival of mice and they have extensively investigated all different aspects of the mechanism of PTX3 protection. One main strength of the manuscript is its usage of bone marrow chimeras in addition to total as well as tissue-specific mouse strains that support their claims.

## Introduction

*Streptococcus pneumoniae* (or pneumococcus) is a Gram-positive extracellular pathogen which colonizes the respiratory mucosa of the upper respiratory tract and represents a major cause of bacterial pneumonia, meningitis, and sepsis in children, elders, and immunodeficient patients. Depending on the virulence factors expressed by the pathogen and on host factors, the disease can evolve to pneumococcal invasive infection, where pneumococcus invades the lower respiratory tract and translocates through the bloodstream into the systemic compartment (*Weiser et al., 2018*).

As a first line of defense against respiratory pathogens, innate immune pattern recognition molecules (PRMs) recognize microbial components and modulate immune response to control infections. Among conserved fluid-phase PRMs, Pentraxin 3 (PTX3), is a member of the pentraxin family characterized by multifunctional properties, including regulation of innate immunity during infections (*Garlanda et al., 2018*). PTX3 is expressed by various hematopoietic and non-hematopoietic cells in response to microbial moieties and inflammatory cytokines (i.e IL-1β and TNF), and it has been associated with the control of various infections by promoting different anti-microbial mechanisms. Indeed, PTX3 participates directly to the elimination of selected microorganisms by promoting phagocytosis, activating the complement cascade and as a component of Neutrophil Extracellular Traps (NET) (*Daigo et al., 2012*; *Jaillon et al., 2014*; *Jaillon et al., 2007*; *Moalli et al., 2010*; *Porte et al., 2019*). Furthermore PTX3 modulates tissue remodeling (*Doni et al., 2015*) and inflammation by tuning complement activation and P-selectin-dependent transmigration (*Deban et al., 2010*; *Lech et al., 2013*), both involved in neutrophil recruitment and in the evolution of respiratory tract infections (*Quinton and Mizgerd, 2015*).

In humans, PTX3 plasma levels increase in the context of inflammation and selected infectious diseases, including pneumococcal pathologies (i.e. community-acquired pneumonia, ventilator-associated pneumonia, pneumococcal exacerbated chronic obstructive pulmonary disease), correlating with the severity of the disease and predicting the risk of mortality (*Bilgin et al., 2018*; *Kao et al., 2013*; *Mauri et al., 2014*; *Porte et al., 2019*; *Saleh et al., 2019*; *Shi et al., 2020*; *Siljan et al., 2019*; *Thulborn et al., 2017*). In addition, single-nucleotide polymorphisms (SNPs) in the *PTX3* gene have been associated with patient susceptibility to respiratory infections (*Brunel et al., 2018*; *Chiarini et al., 2010*; *Cunha et al., 2015*; *Cunha et al., 2014*; *He et al., 2018*; *Olesen et al., 2007*; *Wójtowicz et al., 2015*).

The involvement of PTX3 in the control of selected respiratory pathogens and in the modulation of infection prompted us to investigate the role of this molecule in the control of pneumococcal infections. In a murine model of invasive pneumococcal infection, we observed that PTX3 genetic deficiency was associated with higher disease severity and higher respiratory tract inflammation. PTX3, mainly produced by stromal non-hematopoietic cells during pneumococcal infection, modulated neutrophil recruitment by dampening P-selectin-dependent neutrophil migration. Hence, PTX3 plays a non-redundant role in the control of *S. pneumoniae* infection, modulating neutrophil-associated respiratory tissue damage and pneumococcal systemic dissemination.

## Results

### PTX3 expression during pneumococcal invasive infection

In order to define the relevance of PTX3 in pneumococcal respiratory disease, we first investigated whether the protein is induced during infection. Thus, we used a murine model of pneumococcal invasive infection induced by *S. pneumoniae* serotype 3. Mice were challenged intranasally with $5 \times 10^4$ CFU and sacrificed at different time points. As already described, *S. pneumoniae* serotype 3 causes bacterial colonization of the respiratory tract, then disseminates through the blood circulation and infects other organs like the spleen, resulting in death within 3–4 days (*Figure 1—figure supplement 1A ,B*; *de Porto et al., 2019*). When compared to uninfected mice, infected animals also developed organ dysfunction, as demonstrated by increased circulating levels of creatinine and enzymatic activity of alanine transaminase and creatine phosphokinase 36 hr post-infection (*Figure 1—figure supplement 1C*). As early as 6 hr post-infection, we detected a local expression of PTX3 in the alveolar compartment near the pulmonary veins (*Figure 1A, B*). At 12 hr post-infection, we were able to detect PTX3-specific staining in endothelial cells in the area where we can appreciate inflammatory cells infiltration. This association was confirmed 24 hr post-infection, when a strong PTX3 staining was present

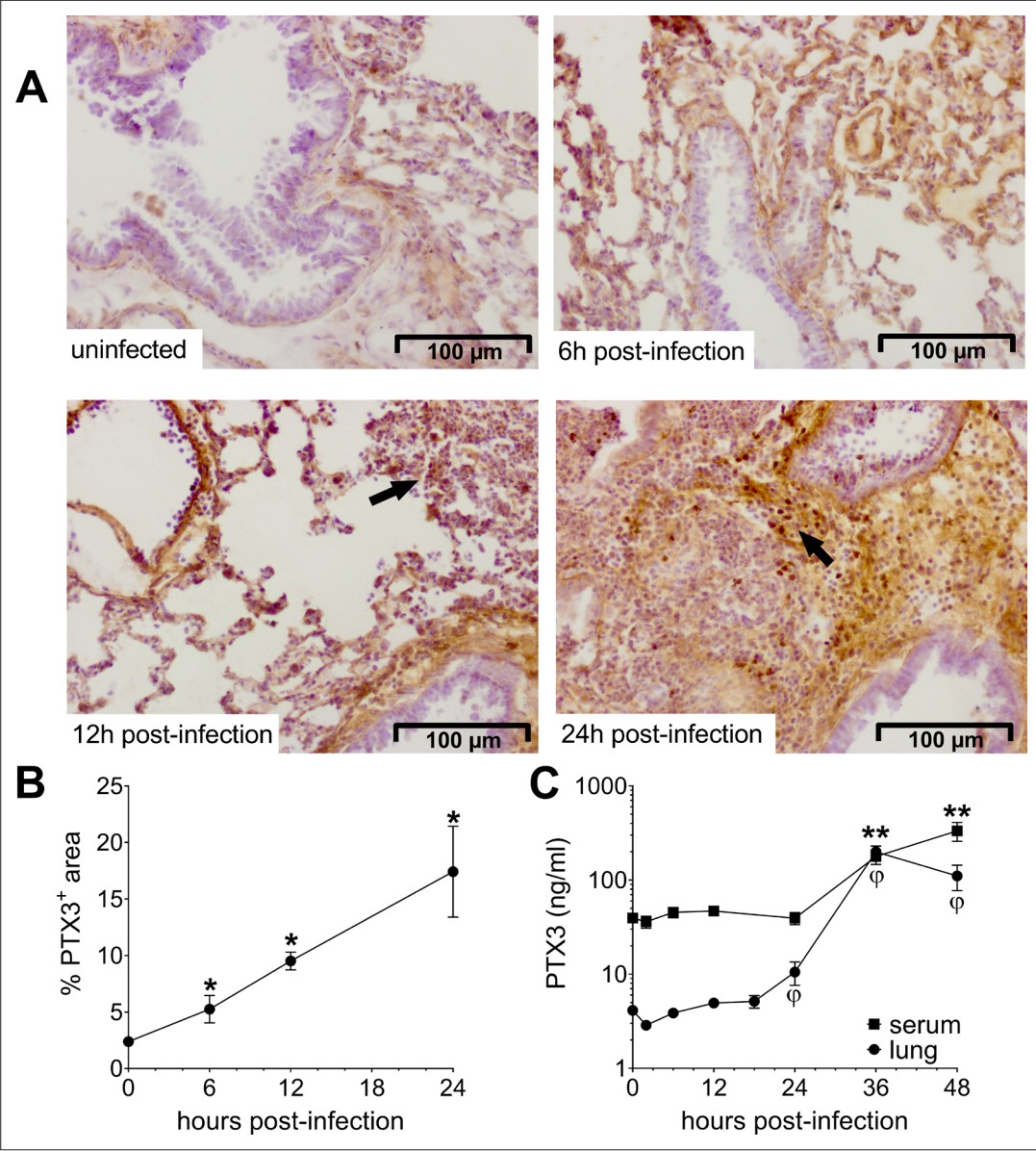

**Figure 1.** Invasive pneumococcal infection induces PTX3 expression. Wild-type (WT) mice were infected intranasally with $5 \times 10^4$ CFU of *S. pneumoniae* serotype 3 and sacrificed at the indicated time points for tissue collection. (**A, B**) Immunohistochemical analysis and quantification of PTX3 expression in lung sections (magnification 20x) from uninfected mice and mice sacrificed 6, 12, and 24 hr post-infection ($n = 3–6$). (**A**) One representative image of at least three biological replicates for each condition is reported. Inflammatory cell infiltrates are indicated by arrows. (**B**) Sections were scanned and analyzed to determine the percentage of PTX3+ area at the indicated time points. (**C**) PTX3 protein levels determined by ELISA in serum and lung homogenates collected at the indicated time points ($n = 4–10$). Results are reported as mean ± standard error of the mean (SEM). Statistical significance was determined using the Mann–Whitney test comparing results to uninfected mice (φ or *p $< 0.05$ and **p $< 0.01$).

The online version of this article includes the following source data and figure supplement(s) for figure 1:

**Source data 1.** Individual data values for the graph in *Figure 1B*.

**Source data 2.** Individual data values for the graph in *Figure 1C*.

**Figure supplement 1.** Bacterial colonization and PTX3 production after infection with *S. pneumoniae* serotype 3.

**Figure supplement 1—source data 1.** Individual data values for the graph in *Figure 1—figure supplement 1A*.

**Figure supplement 1—source data 2.** Individual data values for the survival graph in *Figure 1—figure*

*Figure 1 continued on next page*

*Figure 1 continued*

*supplement 1B*.

**Figure supplement 1—source data 3.** Individual data values for the graph in *Figure 1—figure supplement 1C*.

**Figure supplement 1—source data 4.** Individual data values for the graph in *Figure 1—figure supplement 1D*.

**Figure supplement 1—source data 5.** Individual data values for the graph in *Figure 1—figure supplement 1E*.

near the recruitment site of inflammatory cells forming inflammatory foci (*Figure 1A*). The kinetic of PTX3 production was confirmed by the quantification of PTX3$^+$ area (*Figure 1B*) and by analysis of mRNA in the lung (*Figure 1—figure supplement 1D*). Interestingly, local and systemic production of PTX3 was strongly induced by the infection during the disseminating phase (*Figure 1C*). During this invasive infection we observed that *Ptx3* was upregulated mainly in the lung, aorta, and heart, while other organs like brain, kidneys, and liver did not show higher *Ptx3* expression compared to the uninfected mice (*Figure 1—figure supplement 1E*).

## Induction of PTX3 by IL-1β during *S. pneumoniae* infection

PTX3 has been described to be induced by primary inflammatory cytokines in particular IL-1β (*Garlanda et al., 2018*; *Porte et al., 2019*). In this pneumococcal invasive infection model we observed a rapid induction of IL-1β (*Figure 2A*), and a strong correlation between the levels of IL-1β expressed in the respiratory tract with the levels of lung PTX3 (*Figure 2B*). A similar correlation was observed with TNF and IL-6 levels measured in homogenates from infected lungs (*Figure 2—figure supplement 1A, B*), whereas CXCR1 levels were not correlated with the level of lung PTX3 (*Figure 2—figure supplement 1C*). Moreover, *Il1r*$^{-/-}$ mice infected by *S. pneumoniae* showed lower PTX3 levels, locally and systemically (i.e. in the lung and the serum, respectively) (*Figure 2C, D*). *S. pneumoniae* infected *Myd88*$^{-/-}$ mice were not able to produce PTX3 in the lung and presented the same impairment of PTX3 production as *Il1r*$^{-/-}$ mice (*Figure 2C, D*). These data suggest that, similar to what occurs in other models, PTX3 production during pneumococcal infection requires IL-1β sensing or contribution of MyD88-dependent pathways (*Doni et al., 2015*; *Jaillon et al., 2014*; *Salio et al., 2008*).

## Non-hematopoietic cells are a major source of PTX3 during pneumococcal infection

It has been previously reported that neutrophils contain preformed PTX3, representing an important source of the protein, rapidly released in response to pro-inflammatory cytokines or microbial recognition (*Jaillon et al., 2007*). In agreement, we observed that human neutrophils can release PTX3 upon stimulation with *S. pneumoniae* (*Figure 3—figure supplement 1A*). To investigate the involvement of neutrophils in the production of PTX3 in our model, we used mice lacking granulocyte colony-stimulating factor receptor (*Csf3r*$^{-/-}$). These mice are characterized by chronic neutropenia, granulocyte, and macrophage progenitor cell deficiency and impaired neutrophil mobilization (*Liu et al., 1996*; *Ponzetta et al., 2019*). Following pneumococcal infection, *Csf3r*$^{-/-}$ mice presented lower levels of myeloperoxidase (MPO), a marker of neutrophil recruitment, in lung homogenates at 36 hr post-infection (*Figure 3—figure supplement 1B*). By contrast, even though these mice presented lower amount of neutrophils recruited in response to the infection, they expressed the same pulmonary levels of PTX3 as WT mice (*Figure 3—figure supplement 1B*). These results suggest that neutrophils are not the main source of PTX3 in our murine model of pneumococcal invasive infection.

Since PTX3 can be produced by hematopoietic and non-hematopoietic cells, bone marrow chimeras were used to evaluate the cellular compartment responsible for PTX3 production. During pneumococcal infection, we did not observe any difference in the levels of PTX3 in the respiratory tract and in the serum of WT mice receiving bone marrow from *Ptx3*$^{-/-}$ or WT animals, while no PTX3 was measured in *Ptx3*$^{-/-}$ mice receiving WT or *Ptx3*$^{-/-}$ bone marrow (*Figure 3A, B*). These results suggest that PTX3 is mainly produced by the non-hematopoietic compartment after pneumococcal infection. Endothelial cells were described as an important source of PTX3 (*Garlanda et al., 2018*), thus we evaluated their contribution to PTX3 production during pneumococcal infection. To this aim we crossed conditional *Ptx3*-deficient mice (*Ptx3*$^{Lox/Lox}$) with *Cdh5*$^{Cre/+}$ mice to generate animals with the deletion of PTX3 in endothelial cells. When *Ptx3*$^{Lox/Lox}$*Cdh5*$^{Cre/+}$ mice were infected with *S. pneumoniae*, they presented approximately 50% reduction of PTX3 levels compared to PTX3-competent

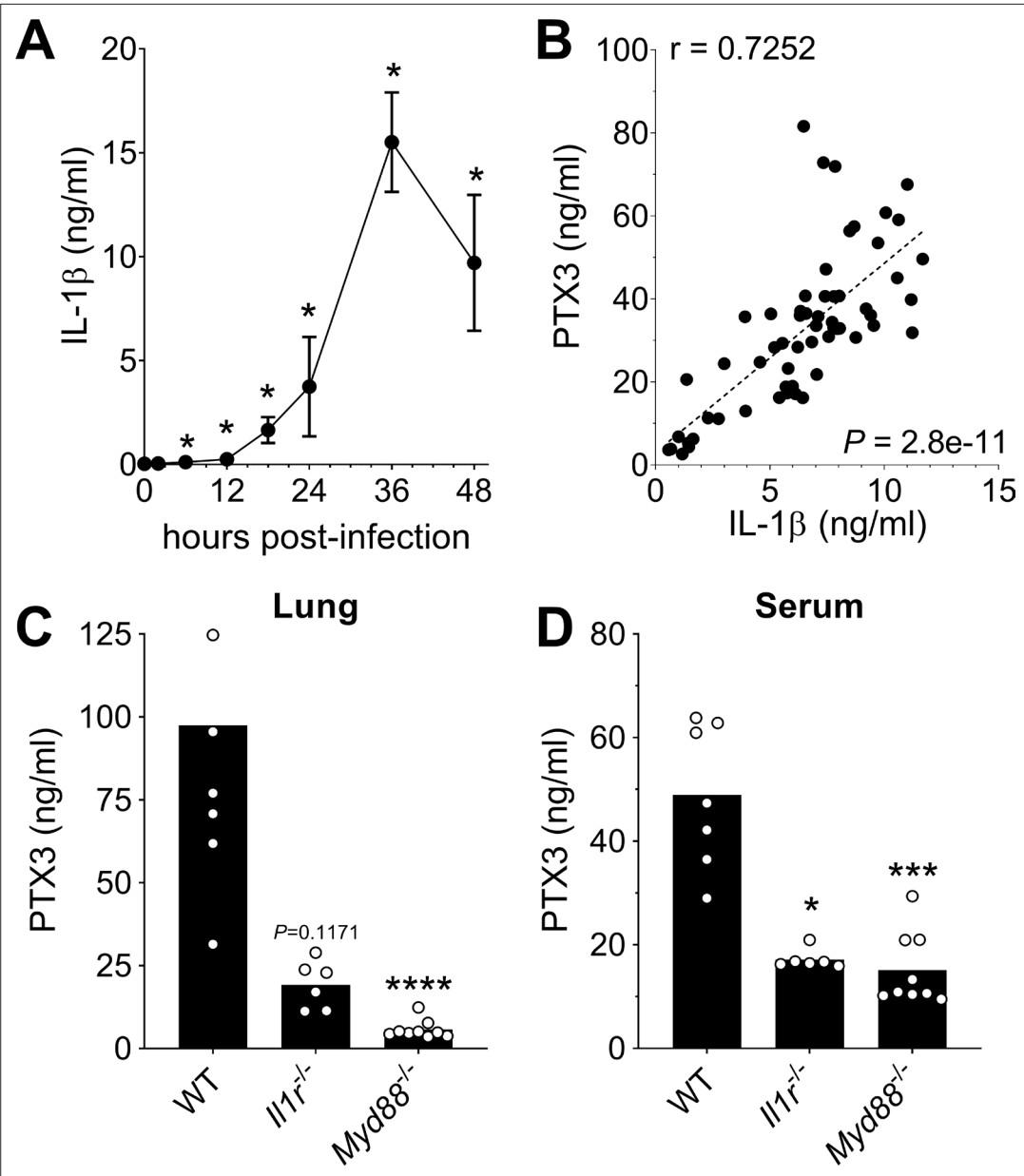

**Figure 2.** Role of IL-1β in induction of PTX3 during *S. pneumoniae* infection. WT mice were infected intranasally with $5 \times 10^4$ CFU of *S. pneumoniae* serotype 3 and sacrificed at the indicated time points for tissue collection. (**A**) IL-1β protein levels in lung homogenates collected at the indicated time points determined by ELISA (*n* = 3–4). (**B**) Correlation between PTX3 and IL-1β protein levels in lung homogenates of all infected mice sacrificed from 2 to 48 hr post-infection (data pooled from five independent experiments, *n* = 60); Pearson correlation coefficient is reported. PTX3 protein levels determined by ELISA in lung homogenates (**C**) and serum (**D**) collected 36 hr post-infection in WT, *Il1r*⁻/⁻ and *Myd88*⁻/⁻ mice (*n* = 7–8). Results are reported as mean ± SEM. Statistical significance was determined using the Mann–Whitney test comparing results to uninfected mice (**A, B**) or the non-parametric Kruskal–Wallis test with post hoc corrected Dunn's test comparing means in *Il1r*⁻/⁻ and *Myd88*⁻/⁻ mice to WT infected mice (**C, D**) (*p < 0.05, ***p < 0.001, and ****p < 0.0001).

The online version of this article includes the following source data and figure supplement(s) for figure 2:

**Source data 1.** Individual data values for the graph in *Figure 2A*.

**Source data 2.** Individual data values for the graph in *Figure 2B*.

**Source data 3.** Individual data values for the graph in *Figure 2C*.

**Source data 4.** Individual data values for the graph in *Figure 2D*.

*Figure 2 continued on next page*

*Figure 2 continued*
**Figure supplement 1.** Correlation of PTX3 levels with pro-inflammatory cytokines induced during *S. pneumoniae* infection.
**Figure supplement 1—source data 1.** Individual data values for the graph in *Figure 2—figure supplement 1A*.
**Figure supplement 1—source data 2.** Individual data values for the graph in *Figure 2—figure supplement 1B*.
**Figure supplement 1—source data 3.** Individual data values for the graph in *Figure 2—figure supplement 1C*.

mice (*Figure 3C, D*). In vitro experiments confirmed the ability of both murine and human endothelial cells to produce PTX3 after stimulation with *S. pneumoniae* (*Figure 3—figure supplement 1C*). Thus, in our setting, non-hematopoietic cells, mainly endothelial cells, are a major source of PTX3.

## Non-redundant role of PTX3 in resistance to pneumococcal infection

Next, we evaluated the role of PTX3 in resistance against pneumococcus. When *Ptx3*−/− mice were infected with *S. pneumoniae* ($5 \times 10^4$ CFU), a significant increase of the bacterial load in the lung was observed during the invasive phase of infection (i.e. 36 hr post-infection), compared to WT mice (*Figure 4A*). Defective local control of bacterial growth was associated to an increase of bacterial load in the spleen (*Figure 4B*). Interestingly there was no difference at earlier time points (i.e. 18 hr post-infection, *Figure 4—figure supplement 1A*), suggesting that PTX3 exerted a role in the control of pneumococcal infection mainly during the invasive phase. Using a bacterial dose ($5 \times 10^3$ CFU) inducing around 30% mortality in WT animals, *Ptx3*−/− mice showed a significant higher mortality (83.3%; p < 0.001) (*Figure 4C*). The phenotype described so far is not restricted to serotype 3 pneumococcus. In fact, when mice were infected with *S. pneumoniae* serotype 1, we observed a strong PTX3 production during the invasive phase of the infection (*Figure 4—figure supplement 1B*) and a correlation with IL-1β levels (*Figure 4—figure supplement 1C*). *Ptx3*−/− mice infected by serotype 1 presented a higher sensitivity to the infection compared to WT animals, with a higher number of bacteria at the local site of infection and also in the spleen 24 hr post-infection (*Figure 4—figure supplement 1D, E*). Thus, in the applied model of *S. pneumoniae* infection, the protection conferred by PTX3 is not limited to serotype 3, and embraces other bacterial serotypes of clinical relevance, including serotype 1.

Systemic administration of recombinant PTX3 to *Ptx3*−/− mice rescues the phenotype. As reported in *Figure 4D*, PTX3 administration in *Ptx3*−/− mice reduced lung colonization to the same level observed in WT mice. We then evaluated the antibacterial activity of PTX3 on *S. pneumoniae* serotype 3. WT animals were treated locally with 1 µg of recombinant protein before infection or 12 hr post-infection. Under both conditions we observed a significant reduction (44% and 57%, respectively; p < 0.01) of the pulmonary bacterial load compared with the CFU found in mice treated with vehicle alone (*Figure 4E*).

## Lack of effective opsonic activity of PTX3

In an effort to explore the mechanism responsible for PTX3-mediated resistance, we first assessed the effect of the recombinant protein on the in vitro growth of *S. pneumoniae*. The incubation of *S. pneumoniae* with 25–250 µg/ml of recombinant PTX3 did not have any effect on the growth rate of the bacteria (*Figure 5—figure supplement 1*).

PTX3 has the capability to act as an opsonin binding selected pathogens and increasing their removal by phagocytosis (*Garlanda et al., 2002*; *Jaillon et al., 2014*; *Moalli et al., 2010*). To assess whether the control of the pneumococcal infection by PTX3 was due to opsonic activity, we first analyzed PTX3 binding to *S. pneumoniae*. By using a flow cytometry assay, we analyzed PTX3 binding to *S. pneumoniae* serotype 3 mimicking the bacteria/PTX3 ratio found in the infected lung ($10^6$ CFU/100 ng PTX3). Under these conditions, we did not observe any interaction of PTX3 with bacteria and, even with an amount of PTX3 5- to 10-fold higher than the one produced in the entire lung, less than 1% of the bacteria were bound (*Figure 5A*). At 500 µg/ml of PTX3 (5000-fold higher than in the lung homogenates) we observed binding to only 36.4% of bacteria (*Figure 5A*).

We then assessed phagocytosis in vitro and in vivo using GFP-expressing *S. pneumoniae* serotype 1 (*S. pneumoniae*-GFP). In a first set of experiments, human neutrophils were incubated with PTX3-opsonized *S. pneumoniae*-GFP. We confirmed that PTX3 exerts opsonic effects, increasing the

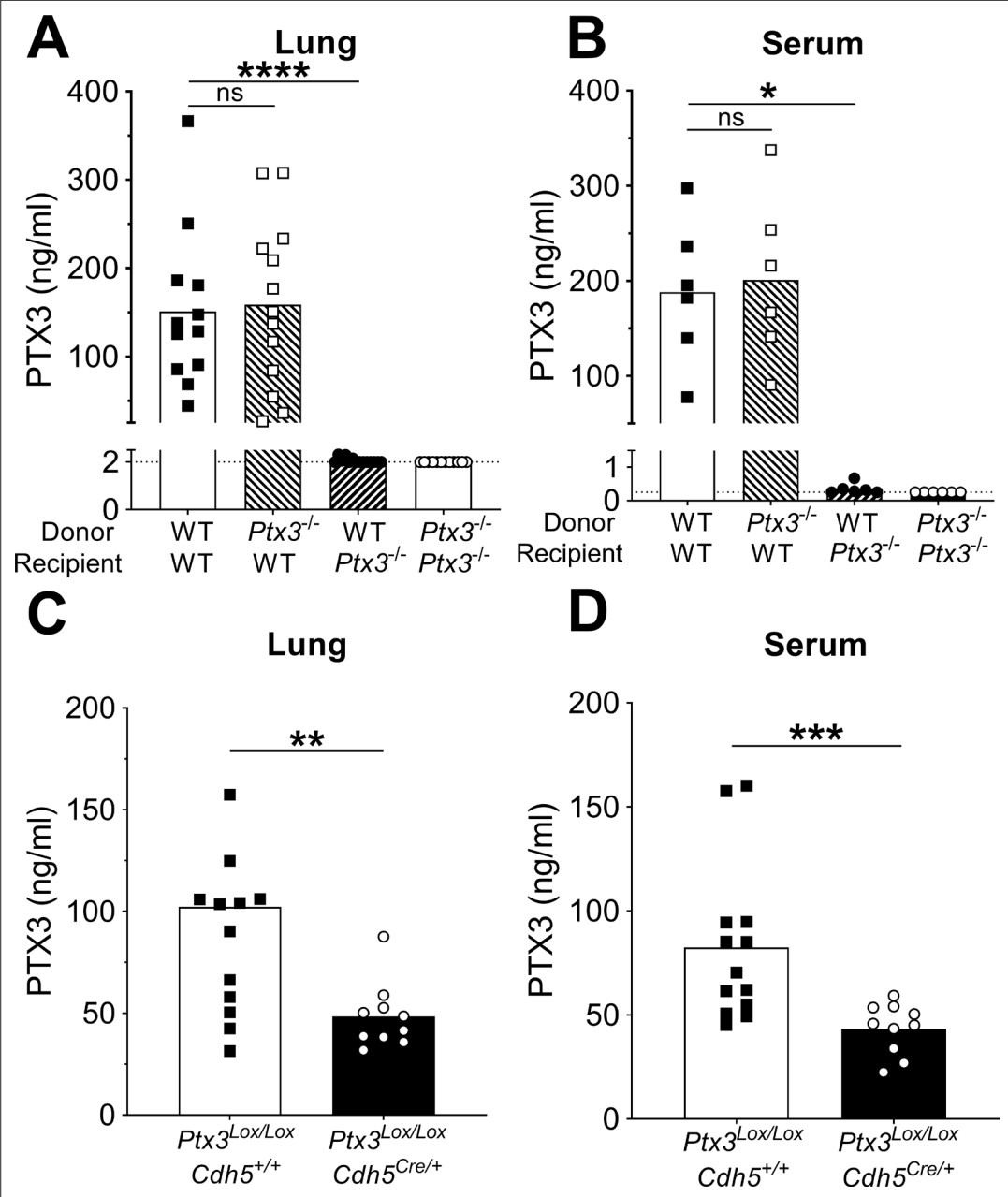

**Figure 3.** Non-hematopoietic cells are a major source of PTX3 during pneumococcal infection. Mice were infected intranasally with $5 \times 10^4$ CFU of *S. pneumoniae* serotype 3 and sacrificed 36 hr post-infection for tissue collection. (**A, B**) PTX3 protein levels determined by ELISA in lung homogenates (*n* = 12–14, **A**) and serum (*n* = 6, **B**) from chimeric mice. Two independent experiments were performed with similar results. (**C, D**) PTX3 protein levels determined by ELISA in lung homogenates (**C**) and serum (**D**) collected from *Ptx3^Lox/Lox^Cdh5^+/+^*, *Ptx3^Lox/Lox^Cdh5^Cre/+^* (*n* = 10–13). Results are reported as mean; PTX3 detection limit is 2 ng/ml in lung homogenates (**A**) and 0.25 ng/ml in serum (**B**) and is represented by a dotted line. Statistical significance was determined using the non-parametric Kruskal–Wallis test with post hoc corrected Dunn's test comparing means to the WT recipient mice reconstituted with WT bone marrow (**A, B**) or the Mann–Whitney test (**C, D**) (*p < 0.05, **p < 0.01, ***p < 0.001, and ****p < 0.0001; ns: not significant).

The online version of this article includes the following source data and figure supplement(s) for figure 3:

**Source data 1.** Individual data values for the graph in *Figure 3A*.

**Source data 2.** Individual data values for the graph in *Figure 3B*.

**Source data 3.** Individual data values for the graph in *Figure 3*.

*Figure 3 continued on next page*

*Figure 3 continued*

**Source data 4.** Individual data values for the graph in *Figure 3D*.

**Figure supplement 1.** Cellular sources of PTX3 in response to stimulation with *S. pneumoniae*.

**Figure supplement 1—source data 1.** Individual data values for the graph in *Figure 3—figure supplement 1A*.

**Figure supplement 1—source data 2.** Individual data values for the graph in *Figure 3—figure supplement 1B*.

**Figure supplement 1—source data 3.** Individual data values for the graph in *Figure 3—figure supplement 1C*.

phagocytosis of pneumococcus by neutrophils, but only at very high concentrations, that is higher than 100 µg/ml (*Figure 5B*). We then moved to an in vivo setting. Since the instillation of as low as 1 µg of PTX3 was sufficient to induce an antibacterial effect when administrated locally just before the infection (*Figure 4E*), we incubated $5 \times 10^4$ CFU of *S. pneumoniae* serotype 3 (i.e. the inoculum normally used for a lethal infection in our model) with 33.3 µg/ml of recombinant PTX3. Mice infected with PTX3-opsonized *S. pneumoniae* serotype 3 showed the same local bacterial burden at 6–36 hr after infection as mice infected with pneumococcus incubated with PBS (*Figure 5C*). We then evaluated the phagocytic ability of neutrophils recruited in vivo during the infection comparing WT and *Ptx3*-deficient mice. Interestingly, we did not observe any difference in the percentage of neutrophils phagocytizing *S. pneumoniae*-GFP in the BAL or in the lung (*Figure 5D*). Finally, we assessed the killing ability of neutrophils collected from WT and *Ptx3*-deficient mice. We did not observe any difference in the percentage of *S. pneumoniae* serotype 3 killed by purified murine neutrophils neither after 1 hr of incubation (WT: 79.02 ± 2.87 and *Ptx3*$^{-/-}$: 80.48 ± 2.87, p = 0.15) or 3 hr of incubation when nearly all pneumococcus were efficiently killed (WT: 99.46 ± 0.30 and *Ptx3*$^{-/-}$: 96.55 ± 5.40, p = 0.31) (*Figure 5E*). These results suggest that the role of PTX3 in resistance to invasive pneumococcus infection is not accounted for by its opsonic activity.

## Regulation of inflammation by PTX3

In pneumococcal invasive disease induced by *S. pneumoniae* serotype 3, infection was characterized by a multifocal neutrophilic bronchopneumonia (*Figure 6—figure supplement 1A*). The main inflammatory cell recruitment was observed during the invasive phase of the infection (starting from 24 hr after infection), when the pulmonary MPO was dramatically increased (*Figure 6—figure supplement 1B*). We analyzed more accurately neutrophil recruitment in the lung and in the BAL of infected mice and we observed two phases of neutrophil recruitment. An initial recruitment, characterized by an increased (i.e. threefold compared to uninfected lung) number of neutrophils both in the BAL and in the lung parenchyma, was observed during the first 6 hr of infection. In the next 12–24 hr of infection we observed an important recruitment of neutrophils in the lung (i.e. 4-fold compared to uninfected lung) that translocated into the alveolar space (up to 50-fold compared to uninfected BAL) (*Figure 6—figure supplement 1C, D*). These two steps of recruitment have been described to exert opposite roles (*Bou Ghanem et al., 2015*). Indeed, the first phase is important for the early control of the infection, reducing the number of colonizing bacteria. In contrast the second phase has been associated with the development of the inflammatory environment, leading to tissue damage that could promote growth and invasion of the bacteria (*Bou Ghanem et al., 2015*). Given the mild expression of PTX3 during the first hours (*Figure 1A*), we investigated the second phase of neutrophil recruitment, comparing *Ptx3*-deficient and WT mice 18 hr after infection. At this time point *Ptx3* deficiency was not associated with a higher respiratory bacterial load (*Figure 4—figure supplement 1A*). Interestingly, the inflammatory profile was significantly increased in *Ptx3*$^{-/-}$ mice, as shown by an increased development of foci in the lung induced by a higher inflammatory cell recruitment (*Figure 6A–C*). Moreover, looking at the time course of the development of pneumococcal-induced respiratory inflammation, we observed that *Ptx3*$^{-/-}$ mice had a quicker and more severe formation of inflammatory foci compared to the WT (*Figure 6B, C*). Furthermore, these mice presented also an increased vascular damage score based on higher perivascular edema and hemorrhages (*Figure 6—figure supplement 1E*). Flow cytometry analysis revealed that the higher inflammation in *Ptx3*-deficient mice was due to a significant increase of neutrophil recruitment in the BAL and the lung (*Figure 6D*). Moreover, we did not observe any change in the recruitment of other myeloid cells, namely macrophages, eosinophils, and

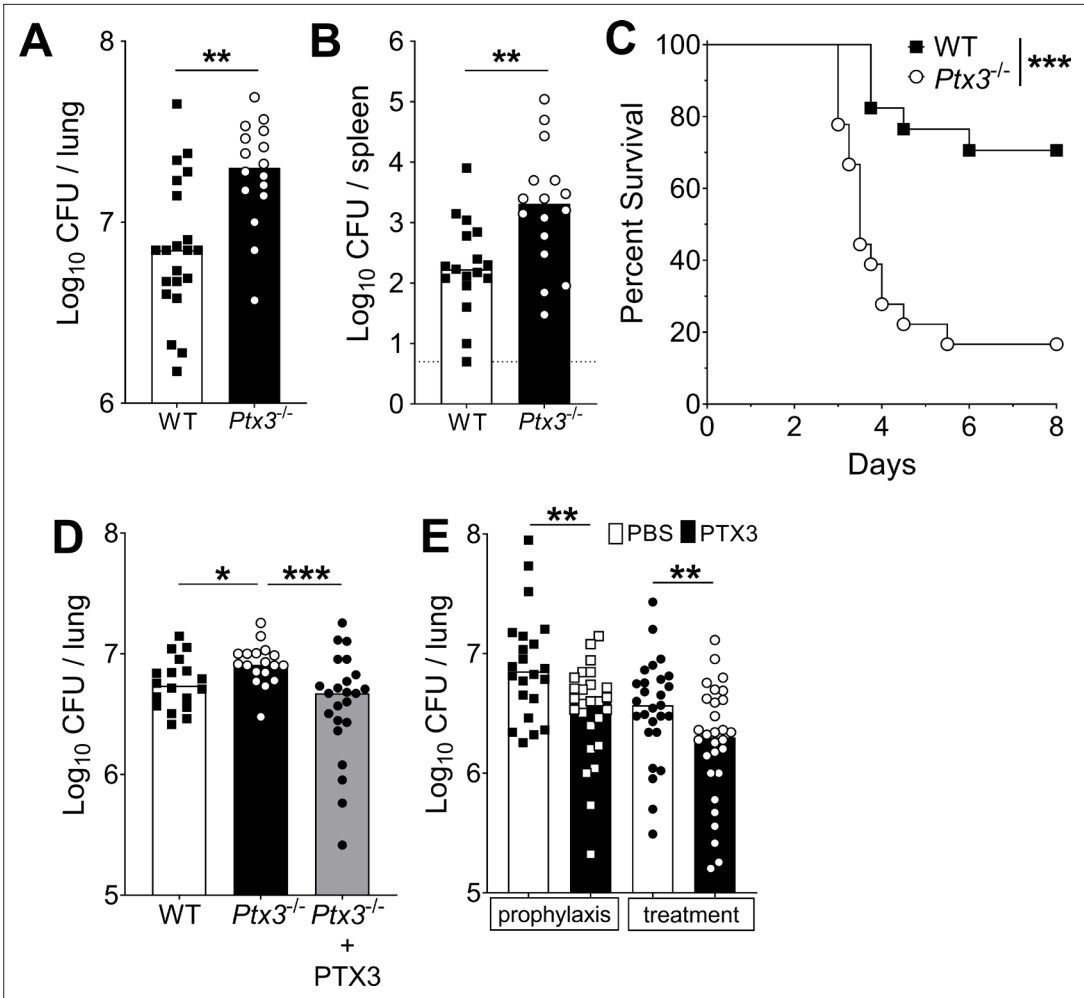

**Figure 4.** Defective resistance of *Ptx3*-deficient mice to invasive pneumococcal infection. WT and *Ptx3*[-/-] mice were infected intranasally with different doses of *S. pneumoniae* serotype 3 and sacrificed at the indicated time points for tissue collection. Bacterial load in lung (**A**) and spleen (**B**) was analyzed at 36 hr post-infection with 5 x 10[4] CFU of bacteria (data pooled from two independent experiments, *n* = 16–21). (**C**) Survival of WT and *Ptx3*[-/-] mice (data pooled from two independent experiments, *n* = 18) was monitored every 6 hr after infection with 5 × 10[3] CFU. (**D**) Bacterial load was analyzed in lungs collected 36 hr post-infection from WT, *Ptx3*[-/-] and *Ptx3*[-/-] mice treated intraperitoneally with recombinant PTX3 (10 μg/100 μl) before the infection and 24 hr post-infection (*n* = 18–23). (**E**) Bacterial load in lungs collected 36 hr post-infection from WT mice treated intranasally before the infection (prophylaxis, data pooled from two independent experiments, *n* = 22–26) or 12 hr post-infection (treatment, data pooled from three independent experiments, *n* = 37–40) with 1 μg/30 μl of recombinant PTX3 or phosphate-buffered saline (PBS). Results are reported as median CFU. Detection limit in the spleen is 5 CFU (dotted line in panel B). Statistical significance was determined using the Mann–Whitney test (**A, B, E**), the non-parametric Kruskal–Wallis test with post hoc corrected Dunn's test comparing means to the WT mice (**D**) and log-rank (Mantel–Cox) test for survival (**C**) (*p < 0.05, **p < 0.01, and ***p < 0.001).

The online version of this article includes the following source data and figure supplement(s) for figure 4:

**Source data 1.** Individual data values for the graph in *Figure 4A*.

**Source data 2.** Individual data values for the graph in *Figure 4B*.

**Source data 3.** Individual data values for the survival graph in *Figure 4C*.

**Source data 4.** Individual data values for the graph in *Figure 4D*.

**Source data 5.** Individual data values for the graph in *Figure 4E*.

**Figure supplement 1.** Infection with *S. pneumoniae* serotype 3 or 1 of WT and *Ptx3*[-/-] mice.

**Figure supplement 1—source data 1.** Individual data values for the graph in *Figure 4—figure supplement 1A*.

*Figure 4 continued on next page*

*Figure 4 continued*

**Figure supplement 1—source data 2.** Individual data values for the graph in *Figure 4—figure supplement 1B*.

**Figure supplement 1—source data 3.** Individual data values for the graph in *Figure 4—figure supplement 1C*.

**Figure supplement 1—source data 4.** Individual data values for the graph in *Figure 4—figure supplement 1D*.

**Figure supplement 1—source data 5.** Individual data values for the graph in *Figure 4—figure supplement 1E*.

monocytes. This phenotype was also observed with the serotype 1 model of invasive pneumococcal infection (*Figure 6—figure supplement 1F*).

Finally, we observed that intranasal treatment with recombinant PTX3 was also associated with a decrease in the neutrophil number in BAL and lungs, demonstrating that PTX3 has a direct role in the control of neutrophil migration in the respiratory tract (*Figure 6E*).

## Regulation of neutrophil recruitment by PTX3 during pneumococcal invasive infection

It has been shown that neutrophil depletion during the invasive phase resulted in protection against pneumococcal infection (*Bou Ghanem et al., 2015*). Accordingly, neutrophil depletion by anti-Ly6G was used to assess the role of these cells in PTX3-mediated protection against pneumococcal infection. In WT mice infected intranasally with *S. pneumoniae*, treatment with anti-Ly6G significantly reduced neutrophils infiltration in the lungs (*Figure 7—figure supplement 1A*). In addition, treatment with anti-Ly6G completely abolished the increased accumulation of neutrophils observed in *Ptx3*⁻/⁻ mice (*Figure 7—figure supplement 1A, B*). The reduction of neutrophil recruitment in both WT and *Ptx3*⁻/⁻ mice treated with anti-Ly6G resulted in a significant reduction of the local and systemic bacterial load, compared to mice treated with the isotype control (*Figure 7A, B*). In addition, *Ptx3*⁻/⁻ mice treated with neutrophil depleting antibody were not more infected than the WT mice (*Figure 7A, B*). These results suggest that taming of pneumococcus-promoting neutrophil recruitment underlies the role of PTX3 in resistance against this bacterial pathogen.

To dissect the mechanism by which PTX3 orchestrates the modulation of inflammation during pneumococcal infection, we first evaluated the level of neutrophil chemoattractants. At 18 hr post-infection, even though there was a higher amount of neutrophils in the airways of *Ptx3*⁻/⁻ mice, we did not detect any differences in the levels of CXCL1 and CXCL2 between *Ptx3*-deficient and WT mice (*Figure 7—figure supplement 1C*). Since PTX3 is a well-known regulator of complement activation (*Haapasalo and Meri, 2019*), we investigated the levels of the two anaphylatoxins C3a and C5a in the lung homogenates of infected mice. No difference in the levels of the potent chemoattractants C3a and C5a, was observed (*Figure 7—figure supplement 1C*). The levels of C3d, a C3 degradation product deposited on the surface of cells and a marker of complement activation in lung homogenates, were similar in *Ptx3*-deficient and WT mice (*Figure 7—figure supplement 1D*).

PTX3 has been described to directly regulate inflammation by binding P-selectin and reducing neutrophil recruitment, dampening rolling on endothelium (*Deban et al., 2010*; *Lech et al., 2013*). We first evaluated P-selectin levels in lungs at steady state and after *S. pneumoniae* infection, excluding the presence of any difference in P-selectin levels both in WT and *Ptx3*-deficient mice (*Figure 7—figure supplement 1E*). Therefore, we investigated whether interaction with P-selectin could be relevant in the regulation of neutrophil recruitment into the lung. We investigated the ability of PTX3 to dampen neutrophil transmigration through endothelial cell layer in vitro, using *S. pneumoniae* as the attractive signal. We observed that PTX3 could block 40% of the neutrophil migration induced by *S. pneumoniae* (*Figure 7C*). Moreover, treatment of endothelial cells with anti-CD62P (anti-P-selectin) antibody induced the same blocking effect. We did not observe any additional blocking effect of PTX3 in association with anti-CD62P, suggesting that PTX3 exerts its blocking effect through P-selectin. To confirm that PTX3 protects infected mice by blocking P-selectin, we used *P-selectin*-deficient mice (*Selp*⁻/⁻). In *Selp*⁻/⁻ mice PTX3 treatment did not reduce the bacterial load (*Figure 7D*). Moreover, we treated WT and *Ptx3*-deficient mice with anti-CD62P, to block P-selectin-dependent neutrophil transmigration during the invasive phase of infection. Anti-CD62P treatment completely abolished the higher neutrophils recruitment in *Ptx3*-deficient mice (*Figure 7—figure supplement 1F, G*). This result suggests that the higher neutrophil infiltration observed during pneumococcal pneumonia in the absence of PTX3 is dependent on P-selectin. Importantly, the reduction of neutrophil recruitment

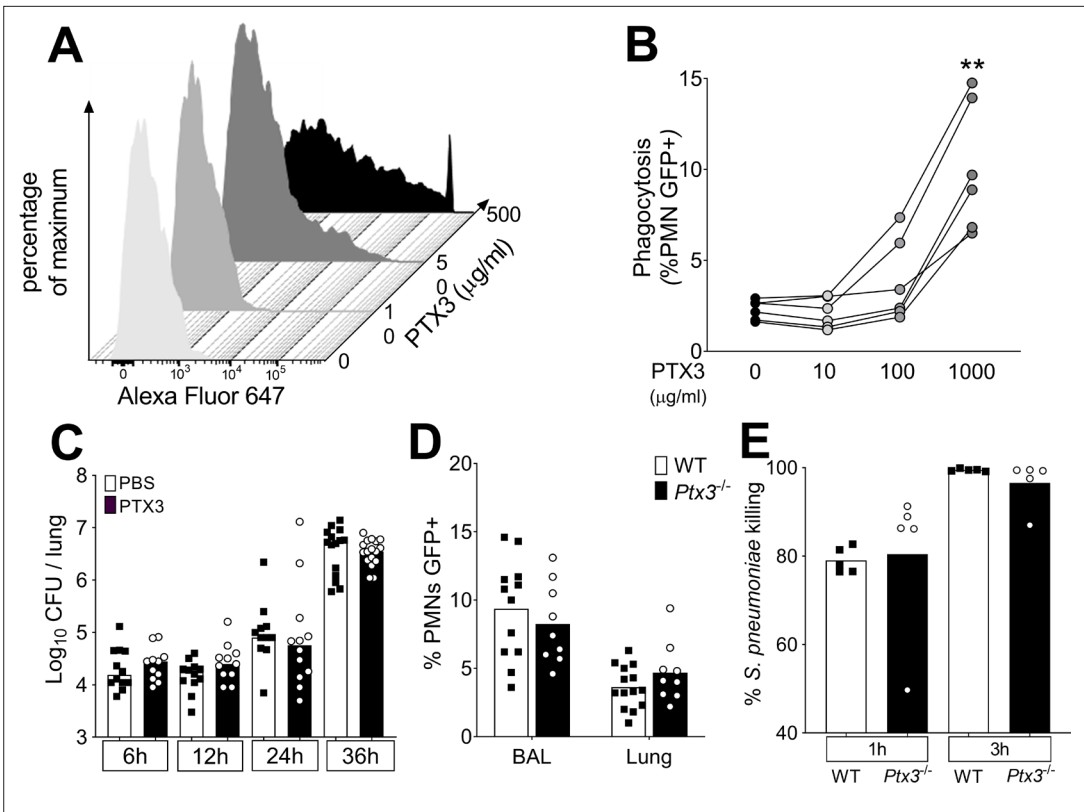

**Figure 5.** Role of phagocytosis in PTX3-mediated resistance to *S. pneumoniae*. (**A**) Binding of biotinylated recombinant PTX3 at the indicated concentration with $10^6$ CFU of *S. pneumoniae* serotype 3 was analyzed by flow cytometry after incubation with Streptavidin-Alexa Fluor 647. (**B**) *S. pneumoniae* serotype 1 expressing GFP (*S pneumoniae*-GFP; $10^6$ CFU) was pre-opsonized with the indicated concentration of recombinant PTX3 and incubated 30 min with $10^5$ purified human neutrophils from six independent donors. GFP-positive neutrophils were analyzed by flow cytometry. Results are expressed as mean of five technical replicates for each time point and donor. (**C**) Bacterial load in lungs collected at indicated time points from WT mice infected intranasally with *S. pneumoniae* serotype 3 pre-opsonized with 33 µg/ml of recombinant PTX3 or non-opsonized (data pooled from two independent experiments, *n* = 11–17). (**D**) Neutrophil phagocytosis of *S. pneumoniae*-GFP was analyzed by flow cytometry. BAL and lungs from WT and *Ptx3*$^{-/-}$ mice were collected 24 hr after infection with a lethal inoculum of *S. pneumoniae* (data pooled from two independent experiments, *n* = 9–14). (**E**) AlamarBlue-based killing assay performed with neutrophils purified from WT and *Ptx3*$^{-/-}$ mice assessed after 1 and 3 hr incubation at a MOI *S. pneumoniae*/neutrophils 2/1. Bars rapresent median values (**C**) or mean values (**D, E**). Statistical significance was determined using the one-way analysis of variance (ANOVA) with Sidak's multiple comparison test (**B**), the non-parametric Kruskal–Wallis test with post hoc corrected Dunn's test comparing means to the WT mice of each time point (**C, E**) and the Mann–Whitney test (**D**) (**p < 0.01).

The online version of this article includes the following source data and figure supplement(s) for figure 5:

**Source data 1.** Individual data values for the graph in *Figure 5B*.

**Source data 2.** Individual data values for the graph in *Figure 5C*.

**Source data 3.** Individual data values for the graph in *Figure 5D*.

**Source data 4.** Individual data values for the graph in *Figure 5E*.

**Figure supplement 1.** Effect of PTX3 on *S. pneumoniae* growth rate.

**Figure supplement 1—source data 1.** Individual data values for the graph in *Figure 5—figure supplement 1*.

**Figure supplement 2.** FACS gating strategy.

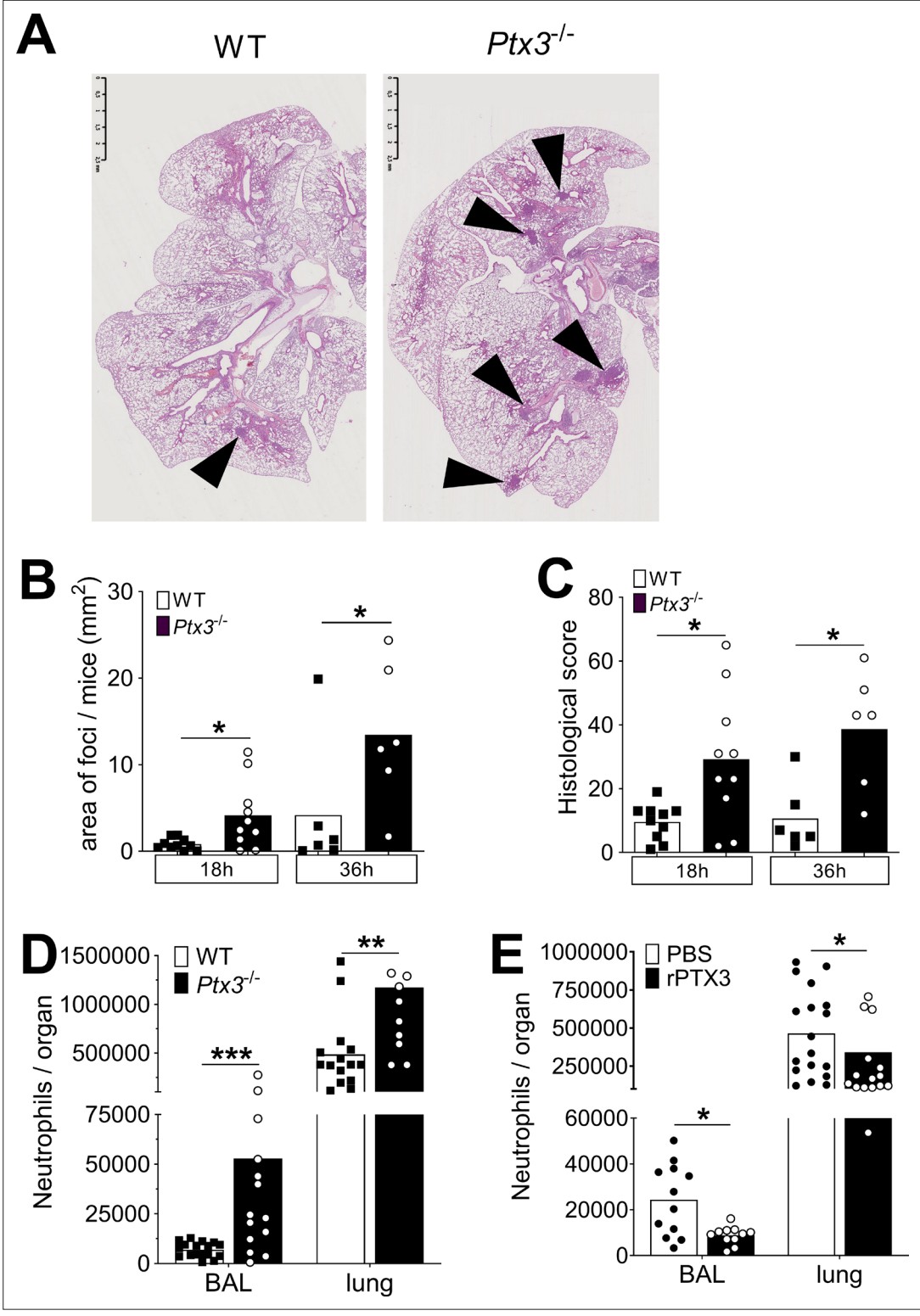

**Figure 6.** PTX3 regulates inflammation during pneumococcus infection. Mice were infected intranasally with 5 × 10⁴ CFU of *S. pneumoniae* serotype 3 and sacrificed at the indicated time points for tissue collection. (**A**) Hematoxylin and eosin (H&E) staining of formalin-fixed histological sections from the lungs of WT and *Ptx3⁻/⁻* mice at 4x magnification. One representative image from at least six biological replicates of WT and *Ptx3⁻/⁻* mice. Inflammatory cell foci are indicated by arrows. (**B**) Area of inflammatory cells foci measured in lungs collected 18 and 36 hr post-infection from WT and *Ptx3⁻/⁻* mice. Areas were measured on three H&E stained lung sections

*Figure 6 continued on next page*

*Figure 6 continued*

per mice at different depth separated by at least 100 µm each (*n* = 6–10). (**C**) Inflammatory histological score measured in lungs collected 18 and 36 hr post-infection from WT and *Ptx3⁻/⁻* mice. Scores (detailed in Materials and methods) were determined on three H&E stained lung sections per mice at different depth separated by at least 100 µm each (*n* = 6–10). (**D**) Neutrophil number determined by flow cytometry in BAL and lungs collected 18 hr post-infection from WT and *Ptx3⁻/⁻* mice (data pooled from two independent experiments, *n* = 11–18). (**E**) Neutrophil number determined by flow cytometry in the BAL and lung collected 18 hr post-infection from WT mice treated intranasally 12 hr post-infection with recombinant PTX3 or PBS (data pooled from two independent experiments, *n* = 11–18). Bars rapresent mean values. Statistical significance was determined using the Mann–Whitney test comparing results to uninfected mice (*$p < 0.05$, **$p<0.01$ and ***$p < 0.001$).

The online version of this article includes the following source data and figure supplement(s) for figure 6:

**Source data 1.** Individual data values for the graph in *Figure 6B*.

**Source data 2.** Individual data values for the graph in *Figure 6C*.

**Source data 3.** Individual data values for the graph in *Figure 6D*.

**Source data 4.** Individual data values for the graph in *Figure 6E*.

**Figure supplement 1.** Neutrophil recruitment during invasive pneumococcus infection.

**Figure supplement 1—source data 1.** Individual data values for the graph in *Figure 6—figure supplement 1B*.

**Figure supplement 1—source data 2.** Individual data values for the graph in *Figure 6—figure supplement 1C*.

**Figure supplement 1—source data 3.** Individual data values for the graph in *Figure 6—figure supplement 1D*.

**Figure supplement 1—source data 4.** Individual data values for the graph in *Figure 6—figure supplement 1E*.

**Figure supplement 1—source data 5.** Individual data values for the graph in *Figure 6—figure supplement 1F*.

in *Ptx3*-deficient mice treated with anti-CD62P is associated with a significant reduction of the local and systemic bacterial load reaching the same level observed in WT mice treated with anti-CD62P (*Figure 7E, F*).

Finally, to assess the role of the P-selectin pathway in PTX3-mediated resistance against invasive pneumococcus infection, we took advantage of *Ptx3⁻/⁻Selp⁻/⁻* double deficient mice. As shown in *Figure 7G, H*, genetic deficiency in P-selectin and PTX3 completely rescued the phenotype observed in *Ptx3⁻/⁻* mice. Thus, the defective control of invasive pneumococcal infection observed in *Ptx3⁻/⁻* mice is due to unleashing P-selectin-dependent recruitment of pneumococcus-promoting neutrophils.

## PTX3 polymorphisms

To explore the significance of these results in humans, we analyzed the association of human *PTX3* gene polymorphisms with invasive pulmonary disease (IPD) in a cohort of 57 patients and 521 age- and sex-matched healthy controls. We focused in particular on two intronic SNPs (rs2305619 and rs1840680) and a third polymorphism (rs3816527) in the coding region of the protein determining an amino acid substitution at position 48 (+734A/C). These SNPs are associated with increased susceptibility to infection to selected microorganisms (*Chiarini et al., 2010*; *Cunha et al., 2014*; *He et al., 2018*; *Olesen et al., 2007*). In addition, the +734A allele was associated in various studies with decreased PTX3 circulating levels (*Barbati et al., 2012*; *Bonacina et al., 2019*; *Cunha et al., 2014*).

Similar frequencies were observed for the +734A allele in patients and controls (67.54% and 61.58%, respectively, p = 0.213, *Table 1*). However, when the haplotypes determined by the three SNPs (rs2305619, rs3816527, and rs1840680) were examined, we found that the AAA haplotype was twice as frequent in IPD patients as in healthy controls (9.67% and 4.26%, respectively, p = 0.0102, *Table 2*). Nevertheless, no significant associations were observed between haplotypes and severity of the disease, evaluated as number of patients developing bacteremia or sepsis (*Table 3*). Frequency was even stronger when considering two SNPs only (+281A and +734A), including the one associated with lower levels of the protein (11.4% and 4.94%, respectively, p = 0.0044, *Table 2*). These observations suggest that also in humans PTX3 could play a role in the control of *S. pneumoniae* infection.

Prompted by the in vivo data, we evaluated P-selectin circulating levels searching for possible association with the different PTX3 haplotypes, however we did not observe any significant association between P-selectin levels and PTX3 haplotypes (*Figure 8—figure supplement 1*).

To assess whether the +734A/C polymorphism in the coding region of the human *PTX3* gene affects the protein's interaction with P-selectin, two recombinant PTX3 constructs were made that carry either

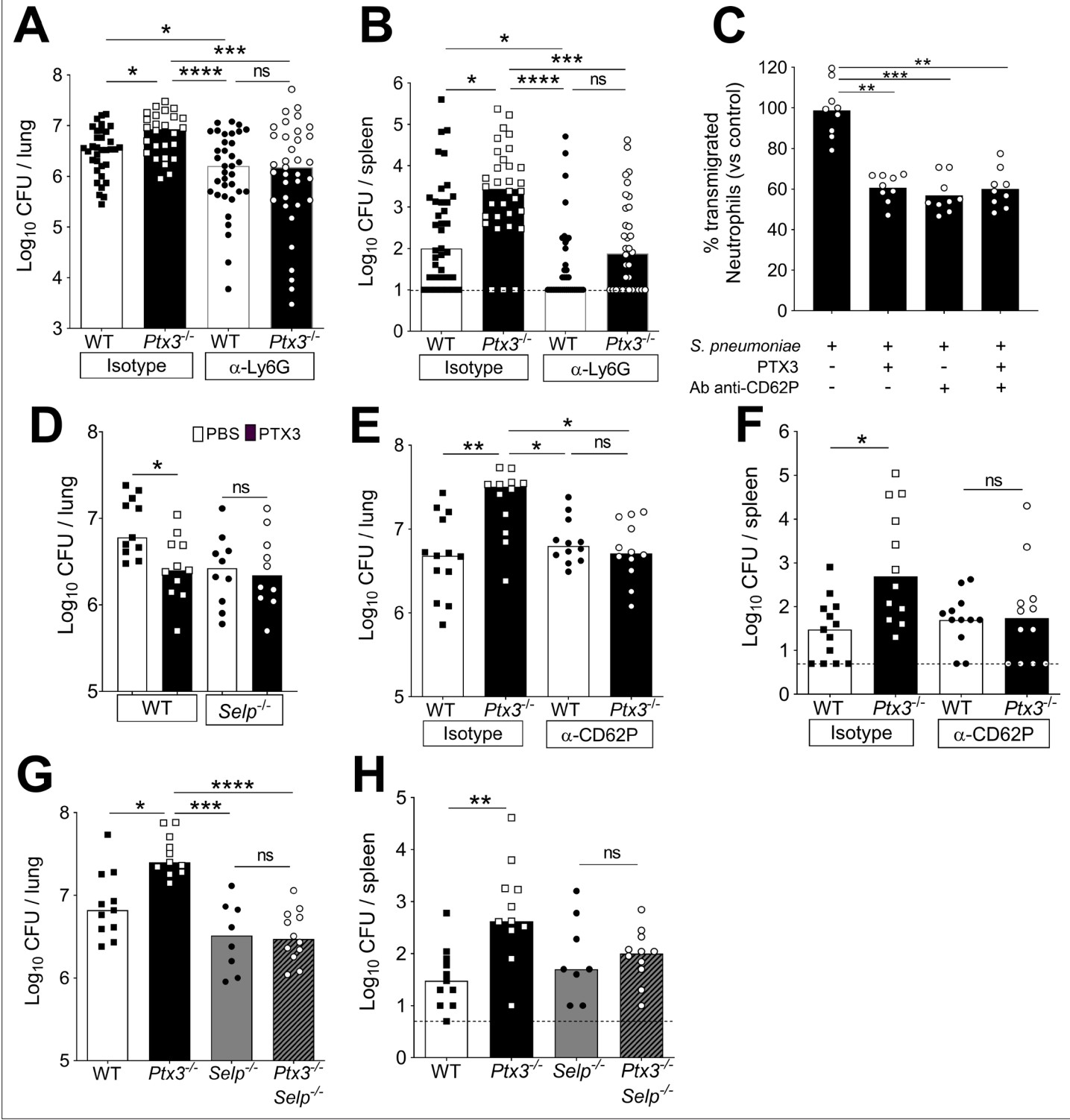

**Figure 7.** Involvement of P-selectin in PTX3-mediated regulation of neutrophil recruitment. Mice were infected intranasally with 5 × 10⁴ CFU of *S. pneumoniae* serotype 3 and sacrificed at the indicated time points for tissue collection. Bacterial load in lung (**A**) and spleen (**B**) collected 36 hr post-infection from WT and *Ptx3⁻/⁻* mice treated intraperitoneally 12 hr post-infection with 200 µg/100 µl of anti-Ly6G or isotype control antibodies (data pooled from three independent experiments, *n* = 26–37). (**C**) Transmigration of human purified neutrophils toward *S. pneumoniae* (data pooled from two independent experiments, *n* = 11). Results are reported as percentage of transmigrated neutrophils considering as 100% the number of transmigrated neutrophils in the control condition (i.e. *S. pneumoniae* in the lower chamber and no treatment in the upper chamber). (**D**) Bacterial load in lungs collected 36 hr post-infection from WT and *Selp⁻/⁻* mice treated intranasally 12 hr post-infection with 1 µg/30 µl of recombinant PTX3

*Figure 7 continued on next page*

*Figure 7 continued*

or phosphate-buffered saline (PBS) (data pooled from two independent experiments, *n* = 10–11). Bacterial load in lungs (**E**) and spleens (**F**) collected 36 hr post-infection from WT and *Ptx3*⁻/⁻ mice treated intravenously 12 hr post-infection with 50 µg/100 µl of anti-CD62P or isotype control antibodies (data pooled from two independent experiments, *n* = 12–13). Bacterial load in lungs (**G**) and spleens (**H**) collected 36 hr post-infection from WT, *Ptx3*⁻/⁻, *Selp*⁻/⁻, and *Ptx3*⁻/⁻*Selp*⁻/⁻ mice (data pooled from two independent experiments, *n* = 8–12). Results are reported as median (**A, B, D–H**) and mean. Detection limits in the spleen is 5 CFU (dotted line in panels B, F, H). Statistical significance was determined using the non-parametric Kruskal–Wallis test with post hoc corrected Dunn's test comparing every means (**A–H**) (*p < 0.05, **p < 0.01, ***p < 0.001, and ****p < 0.0001; ns: not significant).

The online version of this article includes the following source data and figure supplement(s) for figure 7:

**Source data 1.** Individual data values for the graph in *Figure 7A*.

**Source data 2.** Individual data values for the graph in *Figure 7B*.

**Source data 3.** Individual data values for the graph in *Figure 7C*.

**Source data 4.** Individual data values for the graph in *Figure 7D*.

**Source data 5.** Individual data values for the graph in *Figure 7E*.

**Source data 6.** Individual data values for the graph in *Figure 7F*.

**Source data 7.** Individual data values for the graph in *Figure 7G*.

**Source data 8.** Individual data values for the graph in *Figure 7H*.

**Figure supplement 1.** PTX3 modulates neutrophil recruitment in lungs of *S. pneumoniae* infected mice.

**Figure supplement 1—source data 1.** Individual data values for the graph in *Figure 7—figure supplement 1A*.

**Figure supplement 1—source data 2.** Individual data values for the graph in *Figure 7—figure supplement 1B*.

**Figure supplement 1—source data 3.** Individual data values for the graph in *Figure 7—figure supplement 1C*.

**Figure supplement 1—source data 4.** Individual data values for the graph in *Figure 7—figure supplement 1D*.

**Figure supplement 1—source data 5.** Individual data values for the graph in *Figure 7—figure supplement 1E*.

**Figure supplement 1—source data 6.** Individual data values for the graph in *Figure 7—figure supplement 1F*.

**Figure supplement 1—source data 7.** Individual data values for the graph in *Figure 7—figure supplement 1G*.

D (Asp) or A (Ala) at position 48 of the preprotein sequence (corresponding to the A and C alleles of the +734A/C polymorphism, respectively) (*Cunha et al., 2014*). These two proteins had almost identical electrophoretic mobilities when run on denaturing gels both in reducing and non-reducing conditions (*Figure 8A*), where they showed a pattern of bands consistent with previous studies (*Inforzato et al., 2008*). Also, similar chromatograms were recorded when the D48 and A48 variants were resolved on a size exclusion chromatography (SEC) column in native conditions (*Figure 8B*). Given that protein glycosylation is a major determinant of the interaction of PTX3 with P-selectin (*Deban et al., 2010*), it is worth pointing out that the +734A/C polymorphism does not affect structure and composition of the PTX3 oligosaccharides, with major regard to their terminal residues of sialic acid (*Bally et al., 2019*). Therefore, the allelic variants of the PTX3 protein were virtually identical in terms of quaternary structure and glycosidic moiety, which makes them suitable to comparative functional studies. In this regard, when assayed in solid phase binding experiments, these two proteins equally

**Table 1.** Frequency distribution of *PTX3* gene single-nucleotide polymorphisms (SNPs) in invasive pulmonary disease (IPD) patients and controls.

| SNP | Alleles | Amino acid change | Associated allele | Frequency (%) | | $\chi^2$ | OR (95%) | p value |
|-----|---------|-------------------|-------------------|------------------------------|----------------------|----------|-------------------|---------|
| | | | | IPD patient (*n* = 57) | Control (*n* = 521) | | | |
| rs2305619 | +281A/G | | A | 43.86 | 43.44 | 0.007 | 1.02 (0.69–1.50) | 0.931 |
| | | | G | 56.14 | 56.56 | | | |
| rs3816527 | +734C/A | Ala→Asp | C | 32.46 | 38.42 | 1.552 | 0.77 (0.51–1.16) | 0.213 |
| | | | A | 67.54 | 61.58 | | | |
| rs1840680 | +1149A/G | | A | 42.11 | 42.49 | 0.006 | 0.98 (0.67–1.46) | 0.938 |
| | | | G | 57.89 | 57.51 | | | |

**Table 2.** Haplotype analysis for *PTX3* gene in invasive pulmonary disease (IPD) patients and controls.

| rs2305619 | rs3816527 | rs1840680 | Frequency (%) | | $\chi^2$ | p value* |
| | | | IPD patients (*n* = 57) | Controls (*n* = 521) | | |
| --- | --- | --- | --- | --- | --- | --- |
| G | A | G | 56.15 | 56.51 | 0.005 | 0.9409 |
| G | A | | 56.14 | 56.59 | 0.008 | 0.9269 |
| | A | G | 57.89 | 57.41 | 0.010 | 0.9202 |
| A | C | A | 32.44 | 38.28 | 1.488 | 0.2226 |
| A | C | | 32.46 | 38.47 | 1.577 | 0.2091 |
| | C | A | 32.46 | 38.33 | 1.51 | 0.2191 |
| A | A | A | 9.67 | 4.26 | 6.604 | **0.0102** |
| A | A | | 11.4 | 4.94 | 8.129 | **0.0044** |
| | A | A | 9.65 | 4.26 | 6.533 | **0.0106** |
| A | A | G | 1.73 | 0.95 | 0.610 | 0.4348 |

* values in bold are statistically significant

bound plastic-immobilized P-selectin (***Figure 8C***), and C1q (***Figure 8D***, here used as a control), indicating that the +734A/C polymorphism (i.e. the D/A amino acid substitution at position 48 of the PTX3 preprotein) does not affect the interaction of this PRM with P-selectin. It is therefore conceivable that the +734A/C SNP (and the others investigated in our association study) determines reduced expression rather than function of the PTX3 protein in vivo, as observed in other opportunistic infections (***Cunha et al., 2014***).

## Discussion

The general aim of this study was to assess the role of the long pentraxin PTX3 in invasive pneumococcal infection. In a well-characterized murine model of *S. pneumoniae* infection, PTX3 expression was rapidly upregulated in the alveolar and bronchiolar compartments of the lungs as well as in serum. Genetic deficiency of PTX3 was associated with higher inflammation and disease severity, as also indicated by the analysis of *PTX3* genetic variants in a cohort of IPD patients. PTX3 effect was independent from opsonophagocytosis and complement activation. Neutrophil recruitment through interaction with P-selectin was mainly responsible for regulation of tissue damage and pneumococcal systemic dissemination.

Pro-inflammatory cytokines, and in particular IL-1β, are strong inducers of PTX3 in vitro and in vivo (***Garlanda et al., 2018***). In our model, IL-1β and other inflammatory cytokines, namely IL-6 and TNF, were massively produced in response to pneumococcus and correlated with PTX3 circulating levels. In addition, defective PTX3 production after intranasal infection with *S. pneumoniae* was observed in *Il1r*$^{-/-}$ mice, as well as in *Myd88*$^{-/-}$ animals. Similar results were obtained in a model of urinary tract infection (***Jaillon et al., 2014***), or in the context of tissue damage after coronary artery ligation and reperfusion or skin wound-healing (***Doni et al., 2015***; ***Salio et al., 2008***). All together, these data are

**Table 3.** Invasive pulmonary disease (IPD) severity developed by patients and association with *PTX3* haplotypes.

| | AAA | ACA | GAG | AAG |
| --- | --- | --- | --- | --- |
| Bacteremia | 7/9 (77.78%) | 17/27 (62.96%) | 27/43 (65.91%) | 1/1 (100%) |
| Sepsis | 5/9 (55.56%) | 13/27 (48.15%) | 17/42 (37.21%) | 1/1 (100%) |
| Septic shock | 0/7 (0%) | 2/23 (8.7%) | 1/37 (2.70%) | 0/1 (0%) |
| Intensive care | 1/8 (12.5%) | 7/25 (28%) | 7/36 (18.42%) | 0/1 (0%) |

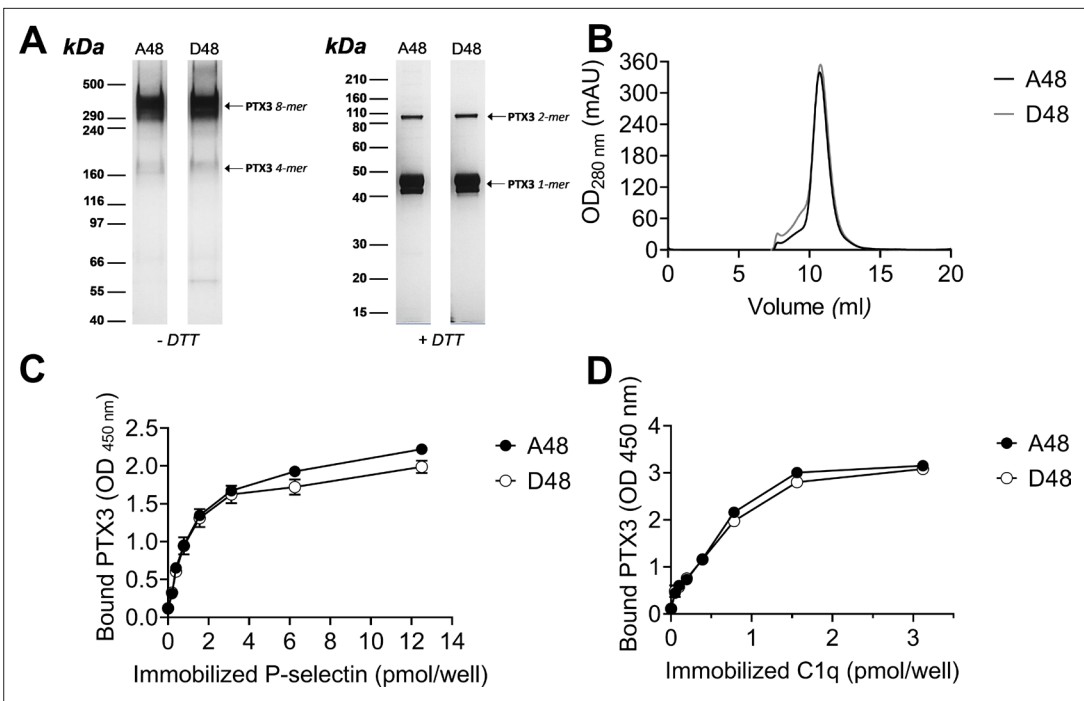

**Figure 8.** Biochemical characterization of the D48 and A48 allelic variants of PTX3 and their binding to P-selectin. (**A**) 500 ng/lane of purified recombinant PTX3 (A48 and D48 from HEK293 cells) were run under denaturing conditions on Tris-Acetate 3–8% (wt/vol) and Bis-Tris 10% (wt/vol) protein gels, in the absence (−) and presence (+), respectively, of dithiothreitol (DTT). Gels are shown with molecular weight markers on the left, and position of the PTX3 monomers, dimers, tetramers, and octamers (1-, 2-, 4-, and 8-mers, respectively) on the right. (**B**) 200 μg aliquots of either one of the two allelic variants were separated on a Superose 6 10/300 GL size exclusion chromatography column in non-denaturing conditions, with elution monitoring by UV absorbance at 280 nm. An overlay of individual chromatograms is presented. (**C, D**) The effect of the +734A/C polymorphism on the interaction of PTX3 with P-selectin was investigated by a solid phase binding assay using microtiter plates coated with the indicated amounts of P-selectin or C1q (here, used as a control) that were incubated with either of the A48 and D48 variants (both at 3 nM). Bound proteins were revealed with an anti-human PTX3 polyclonal antibody, and results are expressed as optical density at 450 nm (OD 450 nm), following background subtraction (n = 3, mean ± standard deviation [SD]). Data shown in panels A to D are representative of three independent experiments with similar results.

The online version of this article includes the following source data and figure supplement(s) for figure 8:

**Source data 1.** Raw images of blot in *Figure 8A*.

**Source data 2.** Individual data values for the graph in *Figure 8B*.

**Source data 3.** Individual data values for the graph in *Figure 8C*.

**Source data 4.** Individual data values for the graph in *Figure 8D*.

**Figure supplement 1.** P-selectin levels in sera from patients with invasive pulmonary disease (IPD).

**Figure supplement 1—source data 1.** Individual data values for the graph in *Figure 8—figure supplement 1*.

consistent with the role of inflammation in the induction of PTX3 and suggest that IL-1β is a major driver of PTX3 production after *S. pneumoniae* infection.

Both myeloid and endothelial cells can produce PTX3 in response to inflammatory cytokines, in addition neutrophils are an important reservoir of the protein. PTX3 is produced during neutrophil maturation in the bone marrow and is stored in specific granules, ready to be released in response to pro-inflammatory signals or upon microbial recognition (*Jaillon et al., 2007*). Despite *S. pneumoniae* ability to stimulate PTX3 release from neutrophils, levels of the protein were similar in WT and neutropenic *Csfr3^−/−* animals infected with *S. pneumoniae*. Bone marrow chimeras and conditional mice definitely demonstrated that stromal cells, and in particular endothelial cells, were a major source of PTX3 in this model of pneumococcal infection. Production of PTX3 by non-hematopoietic cells has been

previously reported in other experimental settings. In a murine model of arterial thrombosis induced by FeCl₃, PTX3 was only expressed by vascular cells (*Bonacina et al., 2016*). Similarly, in a murine model of skin wound-healing, non-hematopoietic cells were the major producers of PTX3 whereas neutrophils showed a minor contribution (*Doni et al., 2015*).

*Ptx3* genetic deficiency was associated with a higher severity to *S. pneumoniae* infection. A defective control of bacterial load, associated with a higher mortality rate, was observed during the invasive phase of the infection, and PTX3 administration rescued the phenotype. In humans, *PTX3* gene polymorphisms were already described to have an impact on the susceptibility to selected infections, induced in particular by *Mycobacterium tuberculosis*, *Pseudomonas aeruginosa* and uropathogenic *Escherichia coli* (*Chiarini et al., 2010*; *Jaillon et al., 2014*; *Olesen et al., 2007*). In addition, *PTX3* genetic variants were associated with the risk to develop invasive aspergillosis or Cytomegalovirus reactivation in patients undergoing allogeneic stem cell transplantation (*Campos et al., 2019*; *Cunha et al., 2014*). In the present study, in a cohort of 57 patients with IPD and 521 healthy controls, haplotypes determined by *PTX3* gene polymorphisms were associated with *S. pneumoniae* infection, albeit no correlation was observed with severity parameters. Thus, genetic deficiency in mice and gene polymorphisms in humans suggest that PTX3 plays an important role in the control of invasive pneumococcal infection.

Various mechanisms are potentially involved in the protective role of PTX3 against infectious agents. PTX3 binds selected fungal, bacterial and viral pathogens, including *Aspergillus fumigatus*, *P. aeruginosa*, *Shigella flexneri*, uropathogenic *E. coli*, Influenza virus, murine cytomegalovirus as well as SARS-CoV-2 nucleocaspid (*Porte et al., 2019*; *Stravalaci et al., 2022*), acting in most cases as an opsonin and amplifying phagocytosis (*Garlanda et al., 2002*; *Jaillon et al., 2014*; *Moalli et al., 2010*). In this study, we observed that PTX3 could bind *S. pneumoniae*, promoting its phagocytosis in vitro by human neutrophils, only at very high concentrations. *Ptx3* deficiency did not affect the local phagocytosis by recruited neutrophils and, given the low efficiency of the binding to pneumococcus, pre-opsonization of the inoculum did not modify the kinetic of infection. Similarly, we did not find any effect of PTX3 in promoting bacterial killing by neutrophils. Phosphatidylcholine is a major constituent of the pneumococcus capsule and a candidate molecule involved in recognition of the microorganism. However, in contrast to the cognate molecule C-reactive protein, that acts as an opsonin for various microorganisms including pneumococcus (*Bottazzi et al., 2010*; *Szalai, 2002*), PTX3 does not bind phosphatidylcholine (*Bottazzi et al., 1997*), likely explaining also the limited binding to *S. pneumoniae*.

PTX3 exerts regulatory roles on complement activation by interacting with components of all the three pathways, that is the classical, alternative, and lectin pathways. In all cases, PTX3 leads to a reduced activation of the complement cascade, thereby reducing the tissue damage associated with an activation out of control (*Haapasalo and Meri, 2019*). However, the higher severity of pneumococcus infection observed in *Ptx3*-deficient mice was not related to failed regulation of complement activity. In fact, similar levels of complement fragments, in particular of the two anaphylatoxins C3a and C5a, were found in lung homogenates of WT and *Ptx3⁻/⁻* infected mice. Overall, PTX3-mediated contribution to resistance to *S. pneumoniae* is independent from enhanced phagocytosis and complement regulation. Indeed, *S. pneumoniae* is characterized by various mechanisms to escape host defense, including production of virulence factors that contribute to complement resistance or the presence of the external capsule rich in phosphatidylcholine (*Andre et al., 2017*; *Weiser et al., 2018*).

The role of neutrophils in invasive pneumococcal infection represents a double-edged sword. In fact, while neutrophil-dependent inflammation could contribute to lung immunopathology in pneumonia (*Palmer and Kimmey, 2022*), other studies have shown a protective effect of neutrophils against pneumococcus respiratory infection by controlling lung damage and reducing neutrophil accumulation and inflammation (*Madouri et al., 2018*; *Porte et al., 2015*; *Tavares et al., 2016*). These results are consistent with a yin/yang role of neutrophils in invasive pneumococcus infection (*Nathan, 2006*).

Several lines of evidence, including depletion using anti-Ly6G antibody, suggested that in early phases of infection neutrophils were an essential component of resistance to *S. pneumoniae* (*Bou Ghanem et al., 2015*). In contrast, neutrophil depletion during the invasive phase was shown to be protective, limiting tissue damage and associated bacterial invasion. In the present study, *Ptx3* genetic deficiency was associated with uncontrolled inflammation and bacterial invasion sustained by enhanced neutrophils accumulation and vascular damages, that could lead to an access for pneumococcus

to nutrients in the alveolar space, allowing its outgrowth and dissemination, as already described (*Sender et al., 2020*). In contrast to our results, a pro-inflammatory role of PTX3 in the context of *S. pneumoniae* serotype 2 infection was reported (*Koh et al., 2017*). However, when we used the same serotype 2 pneumococcus in our setting, we were unable to find any difference in pneumococcal induced inflammatory responses between WT and *Ptx3$^{-/-}$*, suggesting that other factors, including housing conditions, could have influence on the phenotype.

Neutrophil infiltration at sites of bacterial invasion and inflammation is driven by chemoattractants and adhesion molecules (*Maas et al., 2018*). Neutrophil attracting chemokines and complement C5a and C3a were not different in *Ptx3$^{-/-}$* and WT infected mice. Having assessed that neither opsono-phagocytosis nor complement-dependent inflammation were involved in the observed phenotype, we focused on regulation of neutrophils recruitment. In different models of sterile inflammation, such as pleurisy and acid-induced acute respiratory distress syndrome, PTX3 has been shown to serve as a negative regulator of neutrophil recruitment by interacting with P-selectin (*Deban et al., 2010*; *Lech et al., 2013*), thus limiting tissue injury. Despite previous studies in pneumococcus infections reported that neutrophil recruitment was independent of selectins (*Mizgerd et al., 1996*; *Moreland et al., 2004*), in our setting both in vitro studies and in vivo experiments with P-selectin-deficient *Selp$^{-/-}$* mice and *Ptx3$^{-/-}$Selp$^{-/-}$* double deficient mice indicated a defective control of invasive pneumococcal infection in *Ptx3$^{-/-}$* mice. This effect was due to unleashing of P-selectin-dependent recruitment of neutrophils which promote bacterial invasion and tissue damage. On the other hand, we observed that anti-CD62P treatment was only effective in *Ptx3$^{-/-}$* mice, where it reduced neutrophils recruitment in lungs, while no effects were observed in WT mice treated with P-selectin blocking antibody during the early phases of pneumococcal infection. To explain these apparent contrasting results, it is tempting to speculate that in PTX3-competent animals the protein induced by pneumococcal infection could quickly bound P-selectin expressed on endothelial cells, thus explaining the lack of a further response to anti-CD62 treatment.

The results reported here suggest that, by taming uncontrolled P-selectin-dependent recruitment of neutrophils, the fluid-phase PRM PTX3 plays an essential role in tuning inflammation and resistance against invasive pneumococcus infection. A better understanding of the mechanisms of resistance involved in the protection from invasive pneumococcal infection may contribute to the development of novel therapeutic strategies in a personalized medicine perspective.

## Materials and methods

### Ethics statement

Procedures involving animals and their care were conformed with protocols approved by the Humanitas Research Hospital (Milan, Italy) in compliance with national (D.L. N.116, G.U., suppl. 40, 18-2-1992 and N. 26, G.U. March 4, 2014) and international law and policies (EEC Council Directive 2010/63/EU, OJ L 276/33, 22-09-2010; National Institutes of Health Guide for the Care and Use of Laboratory Animals, US National Research Council, 2011). The study was approved by the Italian Ministry of Health (approval n. 742/2016-PR, issued on 26/07/2016). All efforts were made to minimize the number of animals used and their suffering.

Human samples for genotyping of PTX3 SNPs were selected among the patients described by García-Laorden et al., and collected after signature of the informed consent by either the patients or their relatives. Inclusion and exclusion criteria as well as description of the population enrolled were described elsewhere (*García-Laorden et al., 2020*). The investigation conforms to the Helsinki declaration and was approved by the Institutional Review Boards of the participating Hospitals. Peripheral blood neutrophils for in vitro assays were purified from healthy donors upon approval by the Humanitas Research Hospital Ethical Committee. All human samples were anonymized upon collection.

### Mice

All mice used in this study were on a C57BL/6J genetic background. *Ptx3*-deficient mice were generated as described in *Garlanda et al., 2002*. *Ptx3$^{-/-}$* and P-selectin (*Selp$^{-/-}$*) double deficient mice were generated as described in *Doni et al., 2015*. *Csf3r$^{-/-}$* mice were generated as described in *Ponzetta et al., 2019*. Wild-type (WT) mice were obtained from Charles River Laboratories (Calco, Italy) or were cohoused littermates of the gene-deficient mice used in the study. *Ptx3$^{-/-}$*, *Csfr3$^{-/-}$*, *Ptx3$^{Lox/Lox}$Cdh5$^{Cre/+}$*,

$Ptx3^{Lox/Lox}Cdh5^{Cre+/+}$, $Selp^{-/-}$, $Ptx3^{-/-}Selp^{-/-}$, and WT mice were bred and housed in individually ventilated cages in the SPF animal facility of Humanitas Research Hospital or purchased from Charles River Laboratories and acclimated in the local animal facility for at least one weeks prior to infection. All animals were handled in a vertical laminar flow cabinet.

## Bacterial preparation

Each *S. pneumoniae* strain (serotype 1 ST304 and serotype 3 ATCC6303) was cultured and stored as previously described (*Porte et al., 2015*). Briefly, Todd-Hewitt yeast broth (THYB; Sigma-Aldrich) was inoculated with fresh colonies grown in blood agar plates and incubated at 37°C until an optical density at 600 nm ($OD_{600}$) of 0.7–0.9 units was reached. Cultures were stored at −80°C in THYB with 12% glycerol for up to 3 months. GFP-expressing serotype 1 was constructed as described previously (*Kjos et al., 2015*). Clinical isolate E1586 serotype 1 *S. pneumoniae* was grown at 37°C in THYE to an $OD_{600}$ of 0.1, then 100 ng/ml of synthetic competence-stimulating peptide 1 (CSP-1; Eurogentec) was added for 12 min at 37°C to activate the transformation machinery. $P_{hlpA}$-*hlpA-gfp*_Cam$^r$ DNA fragment provided by Jan-Willem Veening's group (*Kjos et al., 2015*) was added to the activated cells and incubated 20 min at 30°C. Growth medium was diluted 10 times with fresh THYB medium and incubated 1.5 hr at 37°C. Transformants were selected by plating 5% sheep blood Tryptic Soy Agar plates (TSA; BD Biosciences) containing 4.5 µg/ml of chloramphenicol, then cultured and stored as described above.

## Mouse model of infection

*S. pneumoniae* serotypes 3 and 1 were used to induce pneumococcal invasive infection as described previously (*de Porto et al., 2019*; *Porte et al., 2015*). For induction of pneumonia, each mouse was anesthetized by intraperitoneal injection of 100 mg/kg of ketamine plus 10 mg/kg of xylazine in 200 µl of PBS. Then $5 \times 10^4$ or $10^6$ colony-forming units (CFU) in 30 µl were inoculated intranasally to induce lethal infection by serotypes 3 and 1, respectively. Mouse survival was recorded every 12 hr. To rescue *Ptx3*-deficient mice, they were treated intraperitoneally with 10 µg/200 µl of recombinant PTX3 prior and 24 hr after infection. Prophylaxis or treatment have been done by intranasal instillation of 1 µg/30 µl recombinant PTX3 prior and 12 hr after infection, respectively. Neutrophil recruitment modulation has been performed by treating intraperitoneally with 200 µg/200 µl of anti-Ly6G depleting antibody (*InVivo*Plus 1A8; BioXcell) or control isotype (*InVivo*Plus rat IgG2a; BioXcell). Blocking of P-selectin was realized by intraperitoneal treatment with 50 µg/100 µl of anti-CD62P depleting antibody (rat RB40.34 NA/LE; BD Biosciences) or control isotype (rat IgG1 $\lambda$; BD Biosciences).

At indicated time, mice were sacrificed with $CO_2$, bronchoalveolar lavage fluid (BAL), serum, lungs, and spleen were harvested and homogenated in PBS for CFU counting or in isotonic buffer (Tris–HCl 50 nM, Ethylenediaminetetraacetic Acid (EDTA) 2 mM, Phenylmethylsulfonyl fluoride (PMSF) 1 mM [Roche Diagnostics GmbH], Triton X-100 1% [Merck Life Science], cOmplete EDTA-free protease inhibitor cocktail [Roche Diagnostics GmbH]) for protein measurement on the supernatant. Bacterial loads per organ were counted by serial dilution plated on 5% sheep blood TSA plates after 12 hr 37°C 5% $CO_2$. Lung CFU were representative of the local infection while splenic CFU were considered as indicator of systemic dissemination of pneumococcus through the bloodstream (*Hommes et al., 2014*; *Porte et al., 2015*; *Schouten et al., 2014*). For histological analysis, the entire lung was collected in organ cassette and fixed overnight in 4% paraformaldehyde (PFA) (immunostaining) or in 10% neutral buffered formalin (hematoxylin–eosin staining).

## Recombinant PTX3

Recombinant human PTX3 was purified from culture supernatant of stably transfected Chinese hamster ovary cells by immunoaffinity as previously described (*Bottazzi et al., 1997*). Purity of the recombinant protein was assessed by sodium dodecyl sulfate–polyacrylamide gel electrophoresis followed by silver staining. Biotinylated PTX3 was obtained following standard protocols. Recombinant PTX3 contained <0.125 endotoxin units/ml as checked by the Limulus amebocyte lysate assay (BioWhittaker, Inc). For in vivo experiments recombinant PTX3 was diluted in PBS.

To assess the effect on the interaction with P-selectin of the rs3816527 (+734A/C) polymorphism in the human PTX3 gene (that results into a D to A amino acid substitution at position 48 of the preprotein), two constructs were generated by overlapping PCR site-directed mutagenesis, and the

corresponding recombinant proteins were expressed in and purified from a HEK293 cell line as previously described (*Cunha et al., 2014*). Aliquots of the purified A48 or D48 PTX3 proteins were run under denaturing conditions on Tris-Acetate 3–8% (wt/vol) and Bis-Tris 10% (wt/vol) protein gels (GE Healthcare Life Sciences), in the absence and presence, respectively, of dithiothreitol as reducing agent. Following separation, protein bands were stained with silver nitrate (ProteoSilver Silver Stain Kit, Sigma-Aldrich). The two recombinant proteins were analyzed in non-denaturing conditions on a Superose 6 10/300 GL SEC column, equilibrated and eluted with PBS at a flow rate of 0.5 ml/min, using an ÄKTA Purifier FPLC system (GE Healthcare Life Sciences). Protein separation and elution were monitored and recorded by UV absorbance at 280 nm.

## Generation of bone marrow chimeras
C57BL/6J WT or *Ptx3*-deficient mice were lethally irradiated with a total dose of 900 cGy. Then, 2 hr later, mice were injected in the retro-orbital plexus with $4 \times 10^6$ nucleated bone marrow cells obtained by flushing of the cavity of a freshly dissected femur from WT or *Ptx3*-deficient donors. Recipient mice received gentamycin (0.8 mg/ml in drinking water) starting 10 days before irradiation and maintained for 2 weeks. At 8 weeks after bone marrow transplantation, animals were infected.

## Lung histology and immunostaining
Immunostaining was performed on 8 µm sections from lungs fixed in 4% PFA, dehydrate in sucrose solution, mounted in OCT embedding compound and stored at −80°C. PTX3 staining was performed as described previously (*Jaillon et al., 2014*). Briefly, sections were stained with 5 µg/ml of rabbit polyclonal antibody anti-human PTX3 as a primary antibody and with MACH 1 universal polymer (Biocare Medical) as a secondary antibody. Staining was revealed with 3,3′Diaminobenzidine (Biocare Medical) and counterstained with hematoxylin and eosin. Slides were scanned and analyzed with Image-pro (Media Cybernetics) to evaluate the percentage of stained area normalized by analyzing the same area for all animals corresponding to about 25% of the section.

Lung histological analysis was performed on formalin-fixed lungs included in paraffin and 3 µm sections were stained with hematoxylin and eosin. A blind analysis was done on three sections per animal distant at least 150 µm and inflammatory foci were measured determining the area of foci and scores. Scores were determined separating small foci (<0.5 mm) and large foci (>0.5 mm) and then calculating as 'Histological score = small foci + large foci × 3'. Vascular damage was scored according to a 5-category scale for perivascular edema and hemorrhage, in which 0 is absent and 1–4 correspond to minimal (or focal), mild (or multifocal, <10% of blood vessels), moderate (or multifocal, 10–50% of blood vessels), and marked (or multifocal, >50% of blood vessels), respectively.

## Binding assay
The binding of PTX3 on *S. pneumoniae* was assessed as described previously (*Bottazzi et al., 2015*). Briefly, $10^6$ CFU *S. pneumoniae* were washed in PBS containing calcium and magnesium (PBS$^{+/+}$) and suspend with 10, 50, or 500 µg/ml of biotinylated recombinant PTX3 for 40 min at room temperature. Bacteria were washed with PBS$^{+/+}$ and stained with streptavidin-Alexa Fluor 647 (4 µg/ml, Invitrogen) for 30 min at 4°C. Washed bacteria were then fixed with 4% formalin for 15 min at 4°C. Bacteria were then read by flow cytometry using FACSCanto II (BD Bioscience). Unstained *S. pneumoniae* were used as negative control.

Binding to P-selectin of the A48 and D48 variants of PTX3 from HEK293 cells was then assessed using 96-well Maxisorp plates (Nunc) coated with a recombinant form of the human P-selectin ectodomain (spanning the 42-771 sequence of the preprotein) commercially available from R&D Systems by adaptation of a published protocol (*Bally et al., 2019*). Purified C1q from human serum (Merck) was used as a control.

## Cell culture and stimulation
Human umbilical vein endothelial cells (HUVEC) were obtained from Lonza (cod LOCC2517, Euroclone, Milan, Italy). Once originally thawed, cells were confirmed CD31$^+$ CD105$^+$ before freezing a stock for subsequent use. HUVEC were grown in 1% gelatin coated wells in M199 medium (Sigma-Aldrich) containing 20% fetal bovine serum (FBS), 100 µg/ml of Endothelial Cell Growth Supplement (ECGS, Sigma-Aldrich), 100 µg/ml of heparin (Veracer; Medic Italia) and 1% penicillin and streptomycin (Pen/

Strep). Murine lung capillary endothelial cells (1G11) were originally isolated by two cycles of immunomagnetic selection of CD31$^+$ cells followed by limiting dilution cloning (*Dong et al., 1997*; *Gonzalvo-Feo et al., 2014*). Expression of CD31, CD34, VCAM-1, ICAM-1, and VE-cadherin was monitored by flow cytometry and resulted stable for at least 40 in vitro passages. 1G11 cells were grown in 1% gelatin coated wells in DMEM with 20% FBS, 100 µg/ml of ECGS, 100 µg/ml of heparin, and 1% Pen/Strep. To evaluate PTX3 expression, human and murine endothelial cells were cultivated to have a confluent monolayer in 12-well culture plates (about 10$^5$ cells/well). Mycoplasma testing was routinely performed on cultured cells by PCR (*Tabatabaei-Qomi et al., 2014*) and only mycoplasma-negative cells were used in all experiments.

Human neutrophils were purified from freshly collected peripheral blood in Lithium Heparin Vacutainer (BD Bioscience) and separated by a two-step gradient separation as previously described by Kremaserova and Nauseef (*Quinn and DeLeo, 2020*). Briefly leukocytes and erythrocytes were separated by a 3% Dextran from *Leuconostoc* spp. (Sigma-Aldrich) sedimentation for 40 min, then leukocytes in the supernatant were separated with Lympholyte-H Cell Separation Media (Cerdalane) and cells from the lower liquid interphase were rinsed with RPMI.

Endothelial cells were stimulated after a wash with the same culture media without Pen/Strep and then incubated with the corresponding medium containing 10$^6$ CFU *S. pneumoniae*, 20 ng/ml recombinant IL-1β (Preprotech) or 100 ng/ml lipopolysaccharide from *E. coli* O55:B5 (Sigma-Aldrich) for 6 hr at 37°C. Cells were then lysate with 300 µl of PureZOL RNA isolation reagent (Bio-Rad). Human neutrophils were stimulated with 10$^7$ CFU/ml of *S. pneumoniae* serotype 3 or 10 ng/ml of phorbol myristate acetate for 6 hr at 37°C; PTX3 released in the supernatant was measured by ELISA, as described below.

## Neutrophil transmigration assay

Neutrophil migration assay across an endothelium monolayer was performed as previously described by Bou Ghanem and collaborators (*Bou Ghanem et al., 2015*). Briefly, basolateral sides of HUVEC monolayer grown 4 days on a 3-µm polyester membrane Transwell (Corning) was infected with *S. pneumoniae* (10$^6$ CFU/ml in RPMI) added to the lower chamber, whereas 100 µl PBS$^{+/+}$ containing 20 ng/ml recombinant IL-1β supplemented with 100 µg/ml PTX3 and/or 100 µg/ml mouse anti-human CD62P (clone AK-4, BD Bioscience) were added to the apical side (upper chamber). After 2.5 hr at 37°C, 5 × 10$^5$ human neutrophils (in 100 µl RPMI) were added to the basolateral side. After 2.5 hr at 37°C, neutrophils in the lower chamber were counted in triplicate. Neutrophil transmigration without infection was performed in parallel as negative control.

## Killing assay

Neutrophil killing of *S. pneumoniae* was evaluated by a resazurin-based cell viability assay using murine purified neutrophils. Briefly, murine neutrophils were purified from bone marrow as previously descried (*Moalli et al., 2010*) and cultivated for 24 hr at 37°C with RPMI containing 10% FBS and 10 ng/ml of granulocyte-macrophage colony stimulating factor (GM-CSF). For the killing assay, 50µl PBS containing 4×10$^5$ CFU *S. pneumoniae* serotype 3 were plated into sterile round bottom 96-wells polypropylene microplates (Corning) and incubated for 1 or 3 hr at 37°C with 2 × 10$^5$ murine purified neutrophils from WT or *Ptx3*$^{-/-}$ mice in the presence of 10% autologous plasma (WT or *Ptx3*$^{-/-}$). After incubation, plates were immediately cooled on ice and supernatant removed after centrifugation at 4°C. *S. pneumoniae* incubated without neutrophils were used as a negative control. Heat killed (60°C, 2 hr) *S. pneumoniae* were considered as positive control in the assay. Neutrophils were then lysated with 200 µl of distilled water with vigorous shaking and remaining *S. pneumoniae* were resuspended in 20 µl RPMI. Preparation of AlamarBlue Cell Viability Reagent and assay were performed according to the manufacturer's instructions (Thermo Fisher Scientific-Invitrogen). A volume of 180 µl AlamarBlue solution (18 µl of AlamarBlue reagent and 162 µl of RPMI) was added to each well. After 4-hr incubation at 37°C, fluorescence (excitation/emission at ≈530–560/590 nm) intensity was measured by microplate reader Synergy H4 (BioTek, France). Results represent ratio of fluorescence intensity values relative to those measured in negative controls.

## Quantitative PCR

Organs homogenated in PureZOL RNA isolation reagent (Bio-Rad) and cell lysate RNAs were extracted with the Direct-zol RNA Miniprep (Zymo Research) and reverse transcribed with the high-capacity cDNA archive kit (Applied Biosystems) following the manufacturer's instructions. cDNA was amplified using the Fast SYBR Green Master Mix on a QuantStudio 7 Flex Real Time PCR Systems (Applied Biosystems). The sequences of primer pairs (Sigma-Aldrich) specific for murine *Gapdh* (Forward, 5′-GCAAAGTGGAGATTGTTGCCAT-3′, Reverse, 5′-CCTTGACTGTGCCGTTGAATTT-3′) and *Ptx3* (Forward, 5′-CGAAATAGACAATGGACTCCATCC-3′, Reverse, 5′-CAGGCGCACGGCGT-3′) were used to evaluate their expression. Relative mRNA levels ($2^{-\Delta\Delta CT}$) were determined by comparing first the PCR cycle thresholds (CT) for *Ptx3* and *Gapdh* (ΔCT), and second, the ΔCT values for the infected/treated and uninfected/untreated (mock/control) groups (ΔΔCT). All amplifications were performed in triplicates.

## Measurement of soluble mediators

Levels of murine C3a, C5a, CXCL1, CXCL2, IL-1β, IL-6, TNF, MPO, PTX3, and P-selectin in lung homogenates and serum were determined by enzyme-linked immunosorbent assay (DuoSet ELISA, R&D Systems and Cloud-Clone corp) following the manufacturer's instructions. Levels of Aspartate transaminase, Alanine transaminase, Creatinine, and Creatine Kinase were measured in the serum of WT mice infected or not with $5 \times 10^4$ CFU of *S. pneumoniae* serotype 3 and sacrificed at 36 hr post-infection using a Beckman Coulter apparatus following the procedures indicated by the manufacturer. Human PTX3 was determined with an in-house ELISA as previously described by Jaillon et al. (*Jaillon et al., 2014*). Briefly, anti-PTX3 monoclonal antibody (100 ng/well, clone MNB4) in 15 mM carbonate buffer (pH 9.6) was coated overnight at 4°C in 96-well ELISA plates (Nunc). Wells were then blocked with 5% dry milk for 2 hr at room temperature. Cell culture supernatants were incubated for 2 hr at room temperature. Biotin-labeled polyclonal rabbit anti-PTX3 antibody (5 ng/ml) was used for the detection and incubated 1 hr at 37°C. Plates were incubated with peroxidase-labeled streptavidin (SB01-61; Biospa) for 1 hr at 37°C. Bound antibodies were revealed using the TMB substrate (Sigma-Aldrich) and 450 nm absorbance values were read with an automatic ELISA reader (VersaMax; Molecular Devices).

## Flow cytometry

BAL fluid samples were obtained after intratracheal injection of 1 ml of PBS supplemented with 5% FBS. Lung cells were isolated after digestion in PBS, supplemented with 20% FBS, 2 mM N-2-hydroxyethylpiperazine-N-2-ethane sulfonic acid (HEPES; Lonza), 100 μg/ml collagenase type IV from *Clostridium histolyticum* (Sigma-Aldrich) and 20 μg/ml of DNAse (Roche Diagnostics GmbH) in C-tubes processed with gentleMACS Octo Dissociator with heaters according to the manufacturer's instructions (Miltenyi Biotec). Lysates were pelleted (500 g 8 min) and red blood cells were lysated with 500 μl of ACK lysing buffer (Lonza) for 5 min. Reaction was stopped with PBS, the cell suspensions were filtered through a 70-μm filter and counted using Türk solution (Sigma-Aldrich). Aliquotes of $10^6$ cells were pelleted by centrifugation (500 g, 8 min), treated with Live/dead fixable aqua (Invitrogen) staining following the manufacturer's instruction and reaction stopped in FACS buffer (PBS, 2% FBS, 2 mM EDTA, 0.05% $NaN_3$). Fc receptors were blocked with anti-mouse CD16/CD32 (20 μg/ml, clone 93; Invitrogen) for 20 min. Cells were stained with an antibody panel able to distinguish macrophages (CD45$^+$, CD11b$^-$, SiglecF$^+$), neutrophils (CD45$^+$, CD11b$^+$, SiglecF$^-$, Ly6C$^+$, Ly6G$^+$), monocytes (CD45$^+$, CD11b$^+$, SiglecF$^-$, Ly6C$^{low/moderate/high}$, Ly6G$^-$), and eosinophils (CD45$^+$, CD11b$^+$, SiglecF$^+$) according to the gating strategy described in *Figure 5—figure supplement 2*. The following antibodies were used: anti-CD45-Brilliant Violet 605 (2 μg/ml, clone 30-F11; BD Bioscience), anti-CD11b APC-Cy7 (1 μg/ml, clone M1/70; BD Bioscience), anti-SiglecF-eFluor 660 (1.2 μg/ml, clone 1RNM44N; Invitrogen), anti-Ly6C-FITC (3 μg/ml, clone AL-21; BD Bioscience), and anti-Ly6G-PE-CF594 (0.4 μg/ml, clone 1A8; BD Bioscience). Flow cytometric analysis was performed on BD LSR Fortessa and analyzed with the BD FACSDiva software.

## Genotyping

DNA was obtained from 57 pediatric patients with IPD and 521 age- and sex-matched healthy controls from the cohort described by *García-Laorden et al., 2020*. The genotyping was performed

as previously described by *Barbati et al., 2012*. Briefly, genomic DNAs extracted from frozen EDTA-whole blood were genotyped by real-time PCR, using TaqMan. In particular, 5 µl samples containing TaqMan Genotyping Master Mix, and specific TaqMan SNP genotyping probes (rs1840680, rs2305619, and rs3816527) were mixed with 20 ng of genomic DNA and genotyped using a Quantstudio 6 Flex System according to the manufacturer's instruction (Applied Biosystems).

## Statistical analysis

Results were expressed as median or mean ± standard error of the mean as indicated. Statistical differences were analyzed using the non-parametric Mann–Whitney test for two groups comparison, or the non-parametric Kruskal–Wallis test with post hoc corrected Dunn's test for multiple comparison of the mean with unequal sample size; survival analysis was performed with the log-rank test with Mantel–Cox method. Correlation coefficients were calculated by Pearson correlation analysis. All the analyses were performed with GraphPad Prism 8.0; p values <0.05 were considered significant.

Sample size estimation was determined for each read-out by performing pilot experiments and determining the Cohen's effect size $d$ (*Lakens, 2013*). Sample size was then estimated using G*Power software (version 3.1.9.7) to perform an a priori power analyses considering the $d$ calculated as described above, an $\alpha$ error probability of 0.05 and 0.01 and a power level (1 − $\beta$ error probability) of 0.8 and considering the appropriated statistical analyses test (*Faul et al., 2007*). Depending on the model, the sample size ranges between 3 and 40. Number of animals used are reported in the appropriate legends to figures.

As for SNP association analyses, these were performed using the PLINK v1.07 program (*Purcell et al., 2007*). All polymorphisms had a call rate of 100%, and were tested for Hardy–Weinberg equilibrium (HWE) in controls before inclusion in the analyses (p-HWE >0.05). In detail, deviations from HWE were tested using the exact test (*Wigginton et al., 2005*) implemented in the PLINK software. For each SNP, a standard case–control analysis using allelic chi-square test was used to provide asymptotic p values, odds ratio, and 95% confidence interval, always referring to the minor allele. Haplotype analysis and phasing were performed considering either all three SNPs together or by using the sliding-window option offered by PLINK. All p values are presented as not corrected; however, in the relevant tables, Bonferroni-corrected thresholds for significance are indicated in the footnote.

## Acknowledgements

This work was partially supported by 'Ricerca Corrente' funding from Italian Ministry of Health to IRCCS Humanitas Research Hospital. In addition, the financial support of Fondazione Cariplo (Contract n° 2015-0564) and Associazione Italiana Ricerca sul Cancro (AIRC – grant IG-2019 Contract n° 23465 and 5x1000 Contract n° 21147) to AM, and of Fondazione Beppe and Nuccy Angiolini to RaPa and AI, are gratefully acknowledged. We also acknowledge Jean-Claude Sirard team 'Bacteria Antibiotics and Immunity', Center for Infection and Immunity of Lille, France, for providing serotype 1 pneumococcal strain, and Tom van der Poll team, Academic Medical Center of Amsterdam, Netherlands, for providing us serotype 3 pneumococcal strain. Boujou to Clement Anfray. CG, FA, BB, and AM are supported by the European Sepsis Academy Horizon 2020 Marie Skłodowska-Curie Action: Innovative Training Network (MSCA-ESA-ITN, grant number 676129). ARG received financial support from Fundação para a Ciência e a Tecnologia (FCT) for PhD grant PD/BD/114138/2016. CT received a scholarship from the Société Académique Vaudoise (Lausanne, Switzerland). We thank the Staff of the Clinical Analysis Laboratory of Humanitas Mater Domini (Varese, Italy) for the support on organ dysfunction analysis.

## Additional information

### Competing interests

Barbara Bottazzi: BB is an inventor of a patent (EP20182181) on PTX3 and obtains royalties on related reagents. Alberto Mantovani: AM is an inventor of a patent (EP20182181) on PTX3 and obtains royalties on related reagents. The other authors declare that no competing interests exist.

## Funding

| Funder | Grant reference number | Author |
|---|---|---|
| Fondazione Cariplo | Contract n° 2015-0564 | Rémi Porte |
| Fondazione AIRC per la ricerca sul cancro ETS | grant IG-2019 Contract n° 23465 | Rémi Porte |
| HORIZON EUROPE Marie Sklodowska-Curie Actions | 676129 | Alberto Mantovani |
| Fundação para a Ciência e a Tecnologia | PD/BD/114138/2016 | Rita Silva-Gomes |
| Ministry of Health - Italy | IRCCS Humanitas Research Hospital | Fatemeh Asgari |
| Fondazione AIRC per la ricerca sul cancro ETS | grant 5x1000 Contract n° 21147 | Rémi Porte |
| Fondazione Beppe and Nuccy Angiolini | | Raffaella Parente Antonio Inforzato |
| Société Académique Vaudoise | Scholarship | Charlotte Theroude |

The funders had no role in study design, data collection, and interpretation, or the decision to submit the work for publication.

## Author contributions

Rémi Porte, Conceptualization, Resources, Data curation, Formal analysis, Supervision, Validation, Investigation, Visualization, Methodology, Writing – original draft, Project administration, Writing – review and editing; Rita Silva-Gomes, Conceptualization, Formal analysis, Supervision, Investigation, Methodology, Writing – original draft, Writing – review and editing; Charlotte Theroude, Raffaella Parente, Camilla Recordati, Formal analysis, Investigation, Writing – review and editing; Fatemeh Asgari, Fabio Pasqualini, Sonia Valentino, Investigation; Marina Sironi, Conceptualization, Supervision, Investigation, Methodology, Writing – review and editing; Rosanna Asselta, Formal analysis, Writing – original draft, Writing – review and editing; Marta Noemi Monari, Data curation, Formal analysis; Andrea Doni, Formal analysis, Investigation, Methodology, Writing – review and editing; Antonio Inforzato, Formal analysis, Supervision, Methodology, Writing – original draft, Writing – review and editing; Carlos Rodriguez-Gallego, Ignacio Obando, Elena Colino, Resources; Barbara Bottazzi, Conceptualization, Formal analysis, Supervision, Funding acquisition, Validation, Visualization, Methodology, Writing – original draft, Project administration, Writing – review and editing; Alberto Mantovani, Conceptualization, Formal analysis, Supervision, Funding acquisition, Visualization, Methodology, Writing – original draft, Project administration, Writing – review and editing

## Author ORCIDs

Rémi Porte http://orcid.org/0000-0001-8311-0202
Ignacio Obando http://orcid.org/0000-0002-4516-1735
Barbara Bottazzi http://orcid.org/0000-0002-1930-9257

## Ethics

DNA was obtained from 57 pediatric patients with invasive pulmonary disease (IPD) and 521 age- and sex-matched healthy controls from the cohort described by Garcia-Laorden and collaborators (García-Laorden et al., 2020). DNA samples were provided by Carlos Rodriguez-Gallego, Ignacio Obando, and Elena Colino.

Procedures involving animals handling and care were conformed to protocols approved by the Humanitas Clinical and Research Center (Rozzano, Milan, Italy) in compliance with national (4D.L. N.116, G.U., suppl. 40, 18-2-1992 and N. 26, G.U. march 4, 2014) and international law and policies (European Economic Community Council Directive 2010/63/EU, OJ L 276/33, 22.09.2010; National Institutes of Health Guide for the Care and Use of Laboratory Animals, U.S. National Research Council, 2011). All efforts were made to minimize the number of animals used and their suffering. The study was approved by the Italian Ministry of Health (742/2016-PR).

Decision letter and Author response
Decision letter https://doi.org/10.7554/eLife.78601.sa1
Author response https://doi.org/10.7554/eLife.78601.sa2

## Additional files

### Supplementary files
• MDAR checklist

### Data availability
All data generated or analyzed during this study are included in the manuscript and supporting file. Source Data files have been provided for Figures 1-8 and supplementary figures.

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
