## [Editor Report]

This submission represents a holistic approach to how pentraxin 3 (PTX3) modulates susceptibility to experimental infection by Streptococcus pneumoniae. The authors have built robust findings on the importance of PTX3 for the survival of mice and they have extensively investigated all different aspects of the mechanism of PTX3 protection. One main strength of the manuscript is its usage of bone marrow chimeras in addition to total as well as tissue-specific mouse strains that support their claims.

---

## [Decision Letter]

**Decision letter after peer review:**

Thank you for submitting your article "Regulation of inflammation and protection against invasive pneumococcal infection by the long pentraxin PTX3" for consideration by *eLife*. Your article has been reviewed by 2 peer reviewers, and the evaluation has been overseen by a Reviewing Editor and Carla Rothlin as the Senior Editor. The following individuals involved in the review of your submission have agreed to reveal their identity: Sebastian Weis (Reviewer #1); Elsa Bou Ghanem (Reviewer #2).

Essential revisions:

– The Introduction is far too long and the reader is missing the aim of the study.

– Can levels of P-selectin be provided in lung homogenates or in animal sera?

– The authors are over-expanding the association of PTX3 with the neutrophil function. However, they need to elaborate if the association that they present at the beginning of their submission with IL-1beta and MyD88 is an epiphenomenon or an independent mechanism that they did not investigate.

– The authors need to better exploit the human study. There is no doubt that the AAA haplotype is more common in patients than in comparators. However, the functional data of association with the animal model are not adequate. Can the authors provide P-selectin data in patients carrying the AAA haplotype? The authors need to run a multivariate analysis including comorbidities to demonstrate that the AAA haplotype is associated with susceptibility to infection or worse outcomes.

– The authors discuss their findings in an ambiguous way and generate questions about the uniqueness of their findings.

– I would like to see further cytokines such as IL-6, TNF, KC from the serum and lung and their correlation to PTX3. Is the correlation specific to IL-1b?

– Does the application of PTX3 as prophylaxis or treatment reduce disease severity in wild-type mice and can it reverse disease severity/mortality in Ptx3 k.o. animals?

– Please add serological data for tissue organ dysfunction to show whether the animals develop multi-organ failure or not.

– in vivo pulmonary neutrophil recruitment in the context of S. penumoniae has been shown in several studies to be independent of P-selectin (Mizger et. al, J. Exp. Med., 1996; Moreland et. al, JI, 2004; Ramos-Sevillano et al., Plos.Path 2016). This is also supported by findings within this manuscript (Figure S6 F and G). So how do the authors reconcile their findings with PTX3 here? This may be an actual mechanism explaining why P-selectin is dispensable for PMN lung recruitment during pneumococcal pneumonia (bound by PTX3 upregulated by infection). A discussion of these points is warranted and is of relevance to the field.

– Do the authors still have samples from experiments presented in Figure 3? If yes assessing MPO levels in the lungs by ELISA as a proxy of PMN recruitment can address the question of how is PTX3 controlling PMN pulmonary recruitment/ if endothelial-derived PTX3 controls PMN recruitment in vivo.

– IRB approvals/information and donor consent pertaining to data from donors (PMN isolation and genotyping) are not listed in the manuscript and need to be added.

---

## [Author Response]

Essential revisions:– The Introduction is far too long and the reader is missing the aim of the study.

We agree with the Reviewers that the length of the Introduction could cause a loss of focus. According to the suggestion, we reduced the Introduction (lines 50-89) in the revised version of the Manuscript and focused on the aim of the study.

– Can levels of P-selectin be provided in lung homogenates or in animal sera?

We thank the Reviewers for raising this important point since it was probably not well described in our manuscript. Given the role of PTX3 binding to P-selectin in the regulation of neutrophil extravasation, we analyzed P-selectin levels in lung homogenate of WT and *Ptx3*^-/-^ mice uninfected or infected with *S. pneumoniae*. The results, showing no differences in P-selectin levels among the different groups, were reported in the original Figure S6E, now Figure 7—figure supplement 1E of the revised version. We modified the description in the Result section (lines 383-385) and in legend to Figure 7—figure supplement 1E (lines 1302-1303) to give more relevance to this piece of information.

– The authors are over-expanding the association of PTX3 with the neutrophil function. However, they need to elaborate if the association that they present at the beginning of their submission with IL-1beta and MyD88 is an epiphenomenon or an independent mechanism that they did not investigate.

We agree with the Reviewers that this is an essential point needing a better clarification. We know that IL1β, as well as IL-1R and MyD88 pathways are involved in the mechanisms determining PTX3 production in response to infections. In particular, the involvement of IL-1R and/or MyD88 pathways in the production of PTX3 has been reported in other contexts, such as in a model of acute myocardial infarction (Salio et al., Circulation 2008), in urinary tract infections (Jaillon et al., Immunity 2014) and in models of tissue repair (Doni et al., JEM 2015). This point has been better explained in the Result section (lines 138-140) and in the Discussion (lines 510-519). The appropriate literature has been cited and one paper not already present in the reference list has been added (Salio et al., Circulation 2008). Concerning the association of PTX3 with neutrophil function, we investigated selected antibacterial mechanisms exerted by neutrophils, namely phagocytosis and bacteria killing, to rule out their importance in the phenotype observed. As shown in Figure 5, none of them seems to play an essential role in our model. On the opposite, we found that the regulation of neutrophil recruitment based on the interaction between PTX3 and P-selectin is essential in *Ptx3^-/-^* animals, which have higher neutrophil infiltration and higher bacterial load (Figure 7G-H). In agreement with this observation, we found that in the absence of P-selectin, PTX3 cannot exert its protective effect (Figure 7D). The Discussion on these data has been modified (lines 597-614)

– The authors need to better exploit the human study. There is no doubt that the AAA haplotype is more common in patients than in comparators. However, the functional data of association with the animal model are not adequate. Can the authors provide P-selectin data in patients carrying the AAA haplotype?

We fully agree with the Reviewers that it would be important to expand human studies. We analyzed by ELISA (R&D cod DY137) P-selectin levels in serum of patients from our cohort carrying AAA, GAG, ACA and GAG haplotypes. Unfortunately, it was possible to have sera only from a limited number of subjects. As shown in the new Figure 8-figure supplement 1, we did not observe any significant difference in the levels of soluble P-selectin comparing AAA haplotype carrying patients to AAG (P<0.99), ACA (P<0.99) or GAG (P=0.83)). However, given the limited number of subjects analyzed, it is not possible to draw final conclusions. This information is described in the Result section (lines 457-460).

The authors need to run a multivariate analysis including comorbidities to demonstrate that the AAA haplotype is associated with susceptibility to infection or worse outcomes.

Few information was available for the patients analysed in this cohort: none of them demises and unfortunately we miss other information on outcomes or clinical scores to be able to run a multivariate analysis.

We could only compare the frequency of sepsis or bacteremia cases in the groups of patient carrying AAA haplotype or the other haplotypes. A slight increase in the frequency of sepsis or bacteremia cases was observed in subjects with the AAA haplotype, but the low number of patients did not allow to reach a statistical significance. We now mention this point as Table 3, in the Result section on lines 446-448 and in the Discussion section (lines 543-544).

– The authors discuss their findings in an ambiguous way and generate questions about the uniqueness of their findings.

We are sorry that our discussion resulted ambiguous, and we thank the Reviewers for underlining this important limit. The discussion has been extensively revised and it now conveys the main messages.

– I would like to see further cytokines such as IL-6, TNF, KC from the serum and lung and their correlation to PTX3. Is the correlation specific to IL-1b?

We measured other cytokines in lung homogenates still available from mouse experiments. A significant correlation was found between PTX3 and IL-6 and TNF but not with CXCL1. These data are shown in the new Figure 2—figure supplement 1A-C and mentioned in the Result section (Lines 131-134) and in the Discussion (Lines 511-513).

– Does the application of PTX3 as prophylaxis or treatment reduce disease severity in wild-type mice and can it reverse disease severity/mortality in Ptx3 k.o. animals?

We agree with the Reviewers that information on the role of PTX3 administration in reduction of mortality would be very interesting. We reported that administration of recombinant PTX3 both as prophylaxis or treatment can reduce disease severity in WT and *Ptx3*^-/-^ animals in terms of CFU reduction (Figure 4D-E). Unfortunately it would be impossible to repeat these experiments analysing the effect on mortality since the Italian rules for animal experimentation are more severe than EU rules (legislative decree n°26 march 4^th^ 2014 – Art. 13), and we would not be allowed to perform further survival experiments in conditions in which we have a surrogate of mortality.

– Please add serological data for tissue organ dysfunction to show whether the animals develop multi-organ failure or not.

We measured molecules associated to organ dysfunction in the serum collected 36 hours post-infection in WT and KO mice and in a group of uninfected mice. We observed a significant increase for ALT (liver disfunction), Creatinin (kidney disfunction) and Creatine Kinase (heart failure) in infected mice compared to uninfected control animals (Figure 1—figure supplement 1C), while we did not observe significant differences in AST levels between control and infected mice. These results are mentioned in lines 99-102. In addition the list of Authors has been updated to acknowledge the contribution of the executor of these tests.

– in vivo pulmonary neutrophil recruitment in the context of S. penumoniae has been shown in several studies to be independent of P-selectin (Mizger et. al, J. Exp. Med., 1996; Moreland et. al, JI, 2004; Ramos-Sevillano et al., Plos.Path 2016). This is also supported by findings within this manuscript (Figure S6 F and G). So how do the authors reconcile their findings with PTX3 here? This may be an actual mechanism explaining why P-selectin is dispensable for PMN lung recruitment during pneumococcal pneumonia (bound by PTX3 upregulated by infection). A discussion of these points is warranted and is of relevance to the field.

We thank the Reviewers for raising this important point since it was not discussed in our manuscript. We believe that in PTX3 competent conditions, P-selectin expressed by endothelial cells is quickly bound by circulating PTX3 induced by pneumococcal infection. This could explain the independence of P-selectin in the neutrophil recruitment during pneumococcus invasive infections described in previous published studies. We added this comment in Discussion section (lines 599-608).

– Do the authors still have samples from experiments presented in Figure 3? If yes assessing MPO levels in the lungs by ELISA as a proxy of PMN recruitment can address the question of how is PTX3 controlling PMN pulmonary recruitment/ if endothelial-derived PTX3 controls PMN recruitment in vivo.

We capitalized on left materials from experiments presented in Figure 3 and we measured MPO levels. MPO levels measured were comparable to those measure in WT animals (Figure 3—figure supplement 1B) and no differences of MPO levels were observed 36 h post-infection between *Ptx3^Lox/Lox^Cdh5^+/+^* and *Ptx3^Lox/Lox^Cdh5^Cre/+^* (Author response image 1).

**Author response image 1. sa2fig1:** Ptx3^Lox/Lox^Cdh5^+/+^ and Ptx3^Lox/Lox^Cdh5^Cre/-^ mice were infected intranasally with 5x10^4^ CFU of S. pneumoniae serotype 3 and sacrificed at 36h post-infection for lung collection. MPO levels were determined by ELISA in lung homogenates. Statistical significance was determined using the non-parametric Mann-Whitney test.

– IRB approvals/information and donor consent pertaining to data from donors (PMN isolation and genotyping) are not listed in the manuscript and need to be added.

We apologize for not including the paragraph on ethical authorizations. An Ethichs statement paragraph is now included in the revised version of the manuscript reporting all the ethical authorization for mouse and human experiments (lines 623-639).